

# Tracking surface ozone responses to clean air interventions under a warming climate in China

Jie Fang[1], Yunjiang Zhang[1*], Didier Hauglustaine[2], Bo Zheng[3], Ming Wang[1], Jingyi Li[1],

Yong Sun[4], Haiwei Li[1], Junfeng Wang[1], Yun Wu[1], Mindong Chen[1], Xinlei Ge[5*]

[1]School of Environmental Science and Engineering, Nanjing University of Information Science and
Technology, Nanjing 210044, China

[2]Laboratoire des Sciences du Climat et de l'Environnement, CNRS-CEA-UVSQ, Université Paris-Saclay,
Gif-sur-Yvette, France

[3]Institute of Environment and Ecology, Tsinghua Shenzhen International Graduate School, Tsinghua
University, Shenzhen 518055, China

[4]State Key Laboratory of Tibetan Plateau Earth System, Environment and Resources, Institute of Tibetan
Plateau Research, Chinese Academy of Sciences, Beijing 100101, China

[5]School of Energy and Environment, Southeast University, Nanjing 210096, China

*Correspondence to*: Yunjiang Zhang (yjzhang@nuist.edu.cn) or Xinlei Ge (xinlei@seu.edu.cn)



**Abstract.** Surface ozone, a major air pollutant with profound implications for human health, ecosystems, and climate, shows long-term trends shaped by both anthropogenic and climatic drivers. Here, we develop a machine learning-based approach – the Fixed Emission Approximation (FEA) – to disentangle the effects of meteorological variability and anthropogenic emissions on summertime ozone trends in China. We identify three distinct phases of ozone trends corresponding to clean air actions. Anthropogenic emissions drove a $+23.2 \pm 1.1$ µg m$^{-3}$ increase in summer maximum daily 8-hour average ozone during 2013–2017, followed by a $-4.6 \pm 1.5$ µg m$^{-3}$ decrease during 2018–2020. However, during 2021–2023, extreme meteorological anomalies – including heatwaves and extended monsoon rainfall – emerged as key drivers of ozone variability. Satellite-derived formaldehyde-to-nitrogen dioxide ratios reveal widespread urban volatile organic compounds-limited regimes, with a shift toward nitrogen oxides-limited sensitivity under influence of heatwaves. Finally, we assess ozone trends under sustained climate warming from 1970 to 2023 based on the FEA framework. The results indicate a significant climate-driven increase in ozone levels across China's urban agglomerations, underscoring the amplifying role of climate change in ozone pollution. Together, these findings highlight the dual influence of anthropogenic and climatic factors on ozone pollution and emphasize the need for integrated strategies that couple emission mitigation with climate adaptation to effectively manage ozone risks in a warming world.



## 1 Introduction


Surface ozone ($O_3$) is a critical air pollutant that poses significant threats to human health (Knowlton
et al., 2004), ecosystems (Agathokleous et al., 2020), and climate (Fishman et al., 1979; Hauglustaine et
al., 1994). It forms through complex photochemical reactions involving nitrogen oxides ($NO_x$) and
volatile organic compounds (VOCs) in the presence of sunlight (Jacob, 2000; Wang et al., 2017)
exhibiting a nonlinear response to its precursors (Guo et al., 2023; Liu and Shi, 2021; Wang et al., 2023a).
Controlling ozone pollution remains a global environmental challenge. In recent years, China has
implemented a series of national clean air actions, most notably the Air Pollution Prevention and Control
Action Plan (2013–2017) and the Three-Year Action Plan for Winning the Blue-Sky War (2018–2020)
(Geng et al., 2024; Zhang et al., 2019; Zheng et al., 2018), that have markedly improved air quality,
particularly by reducing fine particulate matter ($PM_{2.5}$) (Geng et al., 2024; Zhang et al., 2019). However,
surface ozone levels have continued to rise in many regions, raising concerns over the complex drivers
of ozone trends and highlighting the need for scientific attribution to guide effective mitigation strategies
(Li et al., 2019a; Liu et al., 2023; Wang et al., 2023a; Weng et al., 2022).
Long-term ozone variability is jointly influenced by anthropogenic emissions and weather
conditions as well as regional climate (Hallquist et al., 2016; Li et al., 2019b; Wang et al., 2022b). While
emission controls directly regulate precursor abundance, climate change modulates ozone through
chemical feedbacks, meteorological dynamics, and biosphere‐atmosphere interactions (Ma and Yin,
2021; Xue et al., 2020). Over the past century, global surface temperatures have increased by
approximately 1.2 °C relative to the pre-industrial baseline (1850–1900), driven largely by human
activity (Legg, 2021). In a warming world, extreme climate anomalies – such as heatwaves and persistent
rainfall shifts – are expected to intensify (Diffenbaugh et al., 2017). These events are increasingly
recognized as critical modulators of ozone variability through their impacts on photochemistry, vertical
mixing, and precursor transport (Gao et al., 2023; Pu et al., 2017; Wang et al., 2022b).
Quantifying the respective roles of anthropogenic emissions and meteorological variability in
driving ozone trends is therefore essential for evaluating the effectiveness of clean air policies (Li et al.,
2019a; Liu et al., 2023). Previous studies have reported rapid increases in surface ozone concentrations
in key Chinese regions – such as the Beijing–Tianjin–Hebei (BTH) and Yangtze River Delta (YRD) –





during the initial policy phase (2013 – 2017), with increases of approximately 28% and 18%, respectively
(Chen et al., 2020; Li et al., 2019a; Liu et al., 2023). In contrast, a modest decline in ozone levels was
observed during 2018 – 2020, largely attributed to emission reductions (Li et al., 2021; Liu and Wang,
2020b; Wang et al., 2024b; Wang et al., 2023a). However, since 2021, observations indicate a renewed
increase in ozone concentrations (Fig. S1). These fluctuations suggest oscillating trends over the past
decade, the drivers of which remain poorly constrained.

68        Two main approaches have been applied to attribute air pollution trends: chemical transport models

(CTMs) (Li et al., 2021; Liu et al., 2023; Liu and Wang, 2020a) and data-driven statistical frameworks
(Li et al., 2019a; Li et al., 2019b; Li et al., 2020). CTMs simulate atmospheric composition based on
emission inventories, meteorological fields, and chemical mechanisms (Ivatt et al., 2022; Liu and Shi,
2021; Liu et al., 2023; Ye et al., 2024). They allow attribution of trend components to emissions or
meteorology, and can resolve sector-specific impacts. However, these models face challenges, including
uncertainties and temporal lags in emission inventories. Statistical models, on the other hand, rely on
observational datasets and predictor-response relationships without requiring explicit emissions or
chemical schemes (Li et al., 2019a; Li et al., 2019b; Li et al., 2020; Zhai et al., 2019). With the growing
availability of atmospheric big data, statistical and machine learning models have emerged as powerful
tools for trend attribution (Dai et al., 2023; Grange et al., 2018; Vu et al., 2019; Zhang et al., 2025; Zheng
et al., 2023). For instance, Grange et al. (2018) developed a random forest-based framework to isolate
meteorological influences on particulate matter. Similarly, Wang et al. (2023a) used an enhanced
XGBoost model to analyze spatial and temporal ozone patterns in China from 2010 to 2021, confirming
that emission reductions played a key role in recent declines. Other recent efforts have extended such
models to long-term assessments of air pollution drivers under climate change (Wang et al., 2022c).

84        Here, we develop a novel machine learning-based framework – Fixed Emission Approximation

(FEA) – to quantify the respective roles of anthropogenic emissions and meteorological conditions in
shaping summertime surface ozone trends in China. Applying FEA to nationwide observational data from
2013 to 2023, we identify three distinct phases of ozone evolution corresponding to major clean air
actions and policy transitions. We further analyze short-term ozone anomalies associated with extreme
weather events, such as the 2022 heatwave and seasonal monsoon rainfall. To characterize photochemical
regimes, we integrate satellite-derived formaldehyde-to-nitrogen dioxide ($HCHO/NO_2$, FNR) ratios from



Tropospheric Monitoring Instrument (TROPOMI), revealing spatiotemporal shifts in ozone formation
sensitivity across China. Finally, we extend our FEA analysis to evaluate climate-driven ozone trends
from 1970 to 2023, using historical meteorological reanalysis data. Together, these results provide a
comprehensive picture of the anthropogenic and climatic forces shaping surface ozone dynamics in a
rapidly warming and urbanizing China.
**2 Data and Methods**
**2.1 Sampling site and instruments**
Hourly surface ozone concentration data were obtained from the National Environmental
Monitoring Center of China and can be accessed through the open website https://air.cnemc.cn:18007/
(last accessed: May 20, 2024). Hourly meteorological data with a spatial resolution of $0.25° × 0.25°$ were
sourced from the ERA5 reanalysis dataset provided by the European Centre for Medium-Range Weather
Forecasts (ECMWF) and are available for download at https://cds.climate.copernicus.eu (last accessed:
March 20, 2025). For detailed variables, refer to Table S1. The MDA8 ozone TAP dataset (Geng et al.,
2021) for 2013 and 2014 can be downloaded from http://tapdata.org (last accessed: May 20, 2024). The
Tropospheric Monitoring Instrument (TROPOMI) on the Sentinel-5P satellite provides global continuous
observation data for two indicators of $O_3$ precursor substances: nitrogen dioxide ($NO_2$) and formaldehyde
(HCHO) concentrations (Lamsal et al., 2014; Shen et al., 2019). The spatial resolution of TROPOMI
data is 1113.2 meters (approximately $0.009°$ in China) (Ren et al., 2022).
**2.2 Machine learning-based FEA approach**
In this study, we propose a machine learning-based FEA approach to assess the impacts of
meteorological factors and anthropogenic emissions on the year-round variations in ozone concentrations.
First, we construct a regression model using the random forest (RF) algorithm to relate ozone
concentrations to meteorological parameters at various atmospheric heights and to regular emission
surrogate parameters (i.e., time variables). The meteorological parameters include 18 distinct variables
at different altitudes, while the emission surrogate parameters include the month and the hour of the day,
these temporal predictors capture the effects of day-night cycles and workday patterns on air pollutant





concentrations, reflecting the long-term trends in pollutant behavior. The aforementioned variables have
been used as typical emission surrogate input features in previous studies (Grange et al., 2018; Meng et
al., 2025; Shi et al., 2021; Vu et al., 2019). Our modeling strategy involves building and predicting models
for individual cities and for each year from 2015 to 2023. Due to the lack of available observational data
for many cities in 2013 and 2014, we did not develop models for these two years. In our approach, 80%
of the dataset is used for model training, while the remaining 20% is reserved for testing. We perform
ten-fold cross-validation and assess model performance using seven statistical metrics, as listed in Table
S2.

125        Following the construction of the machine learning models for individual cities and years, we

introduce the FEA approach. The key principle of FEA is the assumption that the total emissions of ozone
precursors remain unchanged from the baseline year. Specifically, using the model trained on data from
the baseline year ($i$) as a reference for anthropogenic emissions, we establish hourly-resolution models
for the summer months (June to August) of the baseline year. These models are then applied to predict
ozone concentrations under the meteorological conditions of the prediction year ($j$), while holding the
emission levels constant at those of the baseline year ($i$). The difference between the predicted values and
the observed values for the baseline year ($i$) represents the model residuals ($RES_i$), as shown in Eq. (1).
The difference in observed MDA8 ozone concentrations between two different prediction years ($j_1$, $j_2$) is
driven by the differences in meteorological conditions ($\Delta MET_{i(j1,j2)}$) and anthropogenic emission
controls ($\Delta ANT_{i(j1,j2)}$) (Eq. 2). The term $\Delta MET_{i(j1,j2)}$ represents the changes in meteorological
conditions and can be calculated by the difference between the predicted values, $Pred_{i(j1)}$ and
$Pred_{i(j2)}$, for the corresponding years (Eq. 3). The prediction result $Pred_{i(j)}$ obtained by applying the
model trained with data from year $i$ to the meteorological conditions of year $j$ can be used to calculate
the emission-driving variable $ANT_{i(j)}$ corresponding to the model trained in year $i$ and the
meteorological conditions of year $j$ using Eq. (4). Similarly, the value of $\Delta ANT_{i(j1,j2)}$, representing the
change in anthropogenic emissions between the two years $j_1$ and $j_2$, can be therefore calculated using Eq.
(5). By performing these calculations, we can isolate and quantify the contributions of meteorological
conditions and anthropogenic emission controls to the observed ozone trends. We used a cross-matrix
research method to assess the uncertainty of FEA, with specific formulas available in Supporting Method
S1.



$$OBS_i = Pred_i + RES_i \,, \tag{1}$$

$$\Delta OBS_{(j1,j2)} = \Delta MET_{i(j1,j2)} + \Delta ANT_{i(j1,j2)} \,, \tag{2}$$

$$\Delta MET_{i(j1,j2)} = Pred_{i(j2)} - Pred_{i(j1)} \,, \tag{3}$$

$$ANT_{i(j)} = OBS_j - Pred_{i(j)} - RES_i \,, \tag{4}$$

$$\Delta ANT_{i(j1,j2)} = ANT_{i(j2)} - ANT_{i(j1)} = \left( OBS_{j2} - Pred_{i(j2)} - RES_i \right) - \left( OBS_{j1} - Pred_{i(j1)} - RES_i \right)$$

$$= (OBS_{j2} - OBS_{j1}) - (Pred_{i(j2)} - Pred_{i(j1)}) \,, \tag{5}$$

Model performance was first evaluated through ten-fold cross-validation for the Beijing–Tianjin–
Hebei (BTH) region, revealing high predictive skill between observed and predicted MDA8 ozone levels
during 2015-2023 (Fig. S2). The index of agreement (IOA) ranged from 0.96 to 0.97, with correlation
coefficients ($R$) between 0.93 and 0.95. Root mean square errors (RMSE) and normalized mean bias
(NMB) varied from 16.9 to 21.9 $\mu g\,m^{-3}$ and 8 to 25%, respectively, indicating high model accuracy.
Nationally, the model yielded R values of 0.88–0.91 and IOA of 0.93–0.95, with errors remaining within
acceptable ranges (Tables S3–S8). To assess uncertainty stemming from interannual model training
variability, we applied a matrix-based resampling approach (Supporting Method S1). As shown in Fig.
S3, the relative difference in residuals ranged from -9% to 3%, and remained within $\pm 12\%$ for all
regions – supporting the robustness of the FEA method.
**2.3 Ozone formation regime detection with FNR**
Ozone concentrations show a significant nonlinear relationship with their precursors, which can be
classified into three types: the VOC-controlled zone, the $NO_x$-controlled zone, and the excessive/mixed
zone. The ratio of HCHO to $NO_2$ (FNR) serves as a reactive weighting of $VOC/NO_x$ and is one of the
diagnostic indicators of ozone-sensitive intervals (Sillman, 1995),this is particularly suited to the
analysis of satellite data and has been widely used in related research (Jin et al., 2020; Jin and Holloway,
2015; Wang et al., 2021).Based on the framework described by Ren et al. (2022) and Jin et al. (2015),
we derived a diagnostic approach that is more applicable to our data, and the present study categorizes
ozone sensitivity zones for the summer of 2018-2023 according to the following criteria:
$$FNRavg \ < \ 4.0 \ and \ FNRavg \ + \ FNRsd \ < \ 6.0 : \text{VOC} - \text{controlled zone}$$
$$FNRavg \ > \ 4.0 \ and \ FNRavg \ - \ FNRsd \ > \ 2.0 : \text{NO}_x - \text{controlled zone}$$



173         *Otherwise*: *excessive*/*mixed zones*

174   where *FNRavg* and *FNRsd* denote the time-mean and standard deviation of the FNR for the target

175   time period.

**2.4 FEA-based assessment of climate change impacts on ozone**

177    To further evaluate the long-term impact of climate change on ozone concentrations over China

178   from 1970 to 2023, we extended the framework of our proposed FEA method. The core idea of this

179   analysis is to isolate the influence of long-term meteorological variations on ozone, assuming fixed

180   anthropogenic emissions. Given the availability of relatively complete and continuous hourly ozone

181   observations and meteorological data across China from 2015 to 2023, we selected this period as the

182   basis for constructing emission baselines.

183    Following the modeling protocol described in the section Machine learning-based FEA, we trained

184   nine separate random forest models – each using a different year from 2015 to 2023 as an emissions

185   reference. Inputs included hourly ozone observations, key meteorological predictors, and time-related

186   variables (hour of day and month of year). These trained models were then applied to historical reanalysis

187   meteorology from 1970 to 2023 to simulate ozone trends under constant emissions. This yielded nine

188   independent ozone trajectories, each reflecting the influence of long-term meteorological variability

189   under a different fixed-emissions assumption.

190    While the choice of emission baseline may affect the absolute magnitude of simulated ozone, it does

191   not alter the primary objective: assessing the sensitivity of surface ozone to meteorological drivers over

192   multidecadal timescales (Lecœur et al., 2014; Leung et al., 2018; Wang et al., 2022c). This approach

193   captures the climate-induced ozone signal while adopting the commonly used assumption that emissions

194   are not themselves influenced by climate change – a simplification consistent with prior attribution

195   studies (Dang and Liao, 2019; Leung et al., 2018; Shen et al., 2017; Wang et al., 2022c). For comparison,

196   we also estimated the impact of anthropogenic emission changes on ozone concentrations during the

197   observational window of 2015–2023, using the same FEA methodology and the complete hourly dataset

198   for model training. This dual-track analysis enables a clear distinction between the contributions of

199   climate variability and emission mitigation to observed ozone trends.



**3 Results and Discussion**

**3.1 Spatiotemporal Evolution of Summertime Ozone (2013 – 2023)**

Figure 1 presents the interannual variations in maximum daily 8-hour average (MDA8) ozone concentrations during summertime (June–August) across China, with a focus on five key urban agglomerations: Beijing-Tianjin-Hebei (BTH), Yangtze River Delta (YRD), Fenwei Plain (FWP), Sichuan Basin (SCB), and Pearl River Delta (PRD). From 2013 to 2023, summertime ozone levels displayed distinct temporal patterns across regions, reflecting the impact of successive national emission control phases. During the first phase (2013–2017), nationwide MDA8 ozone increased significantly ($p$ < 0.01), rising from 95.5 to 118.0 $\mu g\,m^{-3}$. This growth was especially pronounced in the BTH and FWP regions, where concentrations increased by 38% and 41%, respectively. In contrast, ozone increases were more modest in the YRD (11%), SCB (15%), and PRD (16%) regions, respectively. These results were consistent with the previous studies (Li et al., 2021; Liu and Wang, 2020a, b; Wang et al., 2023a).

In the second phase (2017–2020), corresponding to the implementation of more stringent emission controls on $NO_x$ and VOCs emissions (Geng et al., 2024; Liu et al., 2023), a moderate national decrease in MDA8 ozone was observed, with concentrations declining to 109.0 $\mu g\,m^{-3}$. The regional declines during this period were most notable in FWP (−16%) and YRD (−15%), while BTH (−6%), SCB (−11%), and PRD (−4%) also showed reductions compared to their concentration peaks observed in 2017. However, this downward trend did not persist. In the third phase (2020–2023), the MDA8 ozone rebounded, reaching 118.4 $\mu g\,m^{-3}$ in 2023 – comparable to its 2017 peak – with a particularly sharp increase during the summer of 2022. From 2021 to 2023, MDA8 ozone concentrations rose by 2.8 $\mu g\,m^{-3}$ in BTH, 3.1 $\mu g\,m^{-3}$ in FWP, 16.1 $\mu g\,m^{-3}$ in YRD, and 18.5 $\mu g\,m^{-3}$ in SCB, respectively.

Figure S1 further illustrates the spatiotemporal evolution of summertime MDA8 ozone across 354 cities in China from 2013 to 2023. On average, 68% of cities exceeded the World Health Organization (WHO) air quality guideline of 100.0 $\mu g\,m^{-3}$ for the MDA8 ozone. Elevated ozone levels were primarily observed in densely populated and economically developed eastern regions, such as North China Plain. Across the five major city clusters, the average ozone levels ranged from 89.4 to 152.8 $\mu g\,m^{-3}$ – substantially exceeding the 43.0 $\mu g\,m^{-3}$ threshold associated with ecosystem productivity loss (Gong et al., 2021), implying significant threats to both human and ecological health. Spatially, ozone hotspot



regions expanded between 2013 and 2017 (Fig. S1 a-e), followed by contraction during 2018-2020 (Fig.

S1 f-i), reflecting initial policy effectiveness. However, this progress stalled from 2021. A sharp reversal

was observed in 2022, with widespread increases in MDA8 ozone (Fig. S1 k), suggesting that the

influence of emerging meteorological extremes or evolving ozone photochemical regimes may be

counteracting the gains from emission reductions.

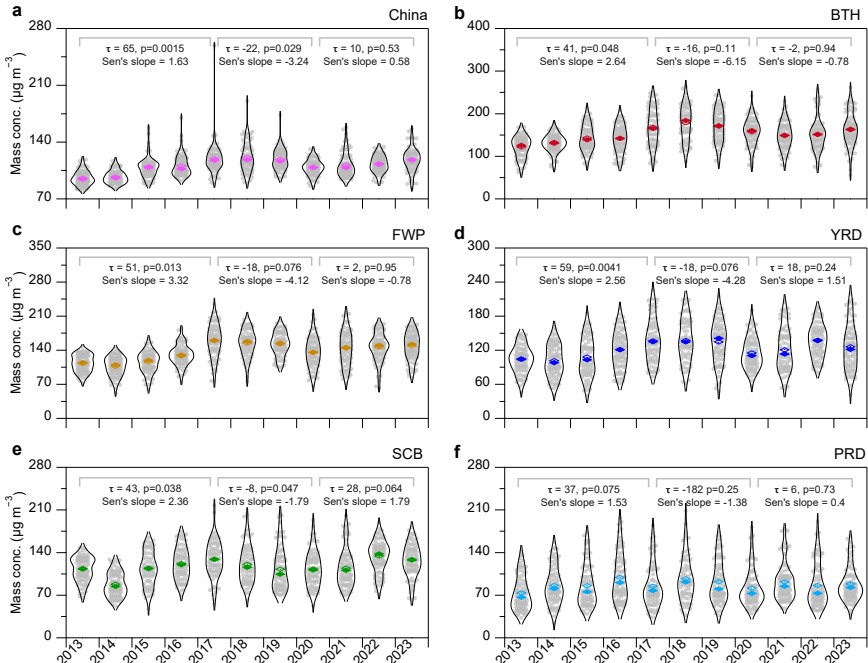

**Figure 1. Interannual trends of summertime MDA8 ozone across China (2013–2023).** Panel (**a**) illustrate the
seasonal variations of MDA8 ozone during the summer months (June, July, and August) across 354 cities nationwide.
Panels (**b-f**) shows the average trend across five key regions in China: Beijing-Tianjin-Hebei (BTH), Fenwei Plain
(FWP), Yangtze River Delta (YRD), Sichuan Basin (SCB), and Pearl River Delta (PRD). The summer months are
defined according to meteorological seasonality, encompassing June, July, and August. In the violin plots, hollow
diamond markers denote the mean, while solid diamond markers represent the median. The Mann-Kendall test and
Sen's slope estimator were employed to assess the statistical significance and rate of change in the monthly average
MDA8 ozone concentrations.

## 3.2 Anthropogenic drivers of ozone trends

To isolate the influence of anthropogenic emissions on summertime ozone variability, we

implemented a machine learning-based FEA framework (Sect. 2.2). This framework employs random

forest (RF) models to disentangle the respective contributions of emission changes and meteorological



variability to observed ozone trends. As illustrated in Fig. 2, anthropogenic emissions were the dominant driver of ozone increases during 2013 – 2017, contributing an average rise of $23.2 \pm 1.1$ µg m$^{-3}$ across 354 cities over China. The strongest regional increases occurred in the FWP and BTH, with contributions of $45.0 \pm 2.0$ µg m$^{-3}$ and $42.1 \pm 2.0$ µg m$^{-3}$, respectively. In contrast, the PRD exhibited a smaller increase ($13.4 \pm 1.6$ µg m$^{-3}$). These findings indicate that the emission control strategies during China's first phase of air quality control efforts, which primarily focused on reducing $PM_{2.5}$ and haze, inadvertently contributed to worsening ozone pollution by altering the atmospheric chemistry and precursor balance (Zhang et al., 2019; Zheng et al., 2018). This finding is consistent with previous model-based assessments using chemical transport models (Li et al., 2021; Wu et al., 2022), and supports the reliability of the FEA framework in attributing observed ozone changes to underlying drivers.

In 2018, China launched its second-phase Clean Air Action Plan, which aimed to the coordinated control of both $PM_{2.5}$ and ozone by reducing $NO_x$ and VOCs emissions (Zhang et al., 2019; Zheng et al., 2018). During this period (2017-2020), summertime ozone concentrations decreased substantially in northern China. As shown in Fig. 2, the MDA8 ozone declined by $10.5 \pm 2.0$ µg m$^{-3}$ in BTH and $10.4 \pm 3.0$ µg m$^{-3}$ in FWP, with smaller but consistent declines in YRD ($-4.8 \pm 3.8$ µg m$^{-3}$), SCB ($-2.8 \pm 2.4$ µg m$^{-3}$), and PRD ($-6.6 \pm 1.4$ µg m$^{-3}$) during 2017 – 2020. These changes underscore the effectiveness of targeted precursor controls and align well with prior studies (Liu et al., 2023; Wang et al., 2023a).

This period also overlapped with the COVID-19 pandemic, which occurred from January to April 2020, introduced an unprecedented, large-scale perturbation to human activity. The nationwide lockdown led to dramatic declines in industrial production, energy consumption, and transportation (Shi and Brasseur, 2020; Zheng et al., 2021). This provided a natural experiment to evaluate the short-term ozone response to abrupt anthropogenic emission reductions. As shown in Fig. S4, from 2017 to 2020, the MDA8 ozone annual mean levels showed a slight national decline, but the pandemic led to an increase in BTH, FWP, YRD, and SCB by +1.7 to +2.3 µg m$^{-3}$, while PRD experienced a decline. Further analysis (Fig. S5) indicates that ~79% of cities saw increases in ozone during this period, with a national average rise of $2.1 \pm 1.3$ µg m$^{-3}$. These increases are consistent with suppressed NO titration and enhanced photochemical ozone production under cleaner atmospheric conditions (Shi et al., 2021; Wang et al., 2022a). In the post-pandemic period (2020–2023), the influence of anthropogenic emissions on

summertime ozone trends became more subdued Emission-driven changes showed relatively small and
mixed contributions across all regions, ranging from –1.2 to +2.6 μg m⁻³ in BTH, –1.6 to +4.0 μg m⁻³ in
FWP, –4.7 to +7.4 μg m⁻³ in YRD, –3.6 to +3.0 μg m⁻³ in SCB, and –3.8 to +7.7 μg m⁻³ in PRD (Fig.
S6). These limited impacts suggest that the benefits of prior emission reduction efforts may have
plateaued, and that other drivers – particularly meteorological extremes – are becoming increasingly
prominent in shaping ozone variability.

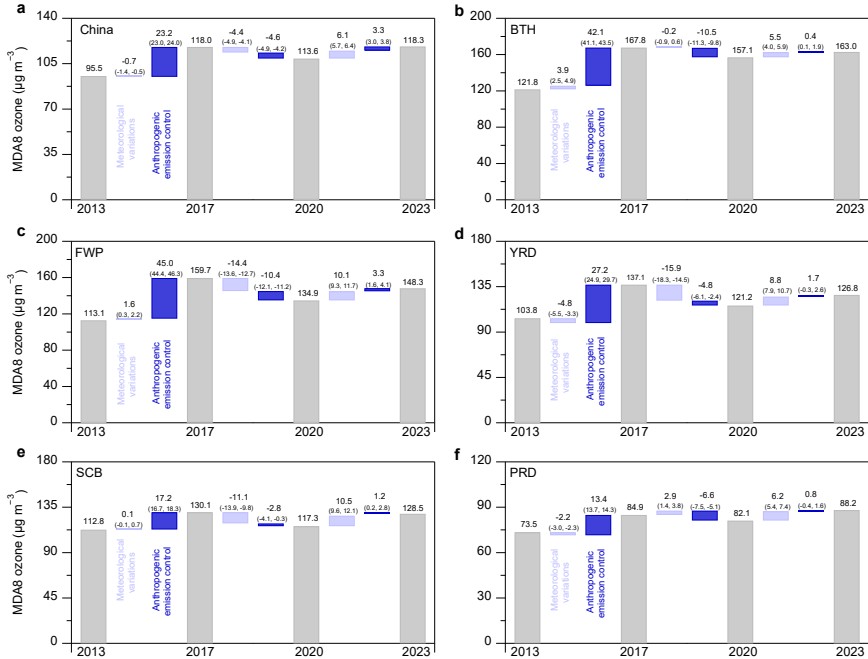


**Figure 2. Anthropogenic and meteorological drivers of ozone trends from 2013 to 2023.** Changes in summerime
MDA8 ozone concentrations were decomposed into contributions from anthropogenic emissions and meteorological
variability using the FEA framework. Results reflect ensemble estimates based on multiple baseline years (2015–
2023) for emissions. Boxplots indicate the interquartile range, with values in parentheses denoting the 25th and 75th
percentiles across all baseline scenarios.

**3.3 Ozone formation sensitivity and regime shifts**
To diagnose the chemical sensitivity of ozone formation, we analyzed the spatial distributions of
tropospheric NO₂ and HCHO columns retrieved by TROPOMI during summer months from 2018 to
2023 (Fig. S7–S8). NO₂ concentrations displayed strong spatial gradients, with eastern China exhibiting
levels five times higher than the west – reflecting dense population centers and elevated anthropogenic



NO$_x$ emissions. While NO$_2$ levels steadily declined over time, the summertime average NO$_2$ column
concentration in the North China Plain decreased from $4.13 \times 10^{15}$ molecules cm$^{-2}$ in 2018 to $3.85 \times 10^{15}$
molecules cm$^{-2}$ in 2023, HCHO concentrations remained relatively stable during 2018–2021. However,
a sharp increase in HCHO was observed in the Yangtze River Delta during the record-breaking heatwave
of 2022, likely due to elevated biogenic and anthropogenic VOC emissions under extreme temperatures
(Qin et al., 2025; Tao et al., 2024). By 2023, HCHO levels returned to near-baseline, consistent with
cooler summer conditions.
To further characterize the photochemical regimes, we derived the threshold of the HCHO/NO$_2$ ratio
(FNR), a widely used proxy for ozone formation sensitivity (Jin and Holloway, 2015; Li et al., 2024; Ren
et al., 2022; Wang et al., 2021). As shown in Fig. 3, extensive VOC-limited and transition zones were
observed in major megacity clusters. most urbanized regions of China remained within the VOC-limited
throughout the study period, with the notable exception of the PRD, which was predominantly NO$_x$-
limited or transitional regimes. This is consistent with previous studies, where VOC-limited regimes
primarily appeared in economically developed and densely populated urban areas, with transition zones
surrounding VOC-limited areas in large cities and suburbs (Li et al., 2024; Shen et al., 2021).
From 2018 to 2020, regime boundaries exhibited only modest changes. During this period, the
VOC-limited areas in the study region gradually decreased, while transition zones correspondingly
increased. Additionally, some areas initially classified as transitional regimes shifted to NO$_x$-limited
regimes. The expansion of mixed and NO$_x$-limited regimes was closely associated with significant NO$_x$
emission reductions (Wang et al., 2023a). In 2021, the VOC-limited area expanded slightly across eastern
China. A more dramatic shift occurred in 2022, as extreme heat and elevated VOC levels drove
widespread transitions from VOC-limited to transitional or NO$_x$-limited regimes, especially across the
YRD and surrounding regions.
Monthly regime evolution from 2020 to 2023 (Fig. S9) confirms that the most extensive regime
shifts occurred in August 2022 (Fig. S9i), coinciding with peak temperatures and FNR anomalies.
Notably, VOC-limited areas tended to be smaller in July and August compared to June, likely due to
increased VOC reactivity under higher temperatures (Fig. S9). However, major cities generally remained
VOC-limited, while adjacent suburban areas shifted dynamically between transitional and VOC-limited
regimes. In contrast, outer suburbs and rural regions were more frequently controlled by NO$_x$ (Shen et



al., 2021; Wang et al., 2021). Although VOC-limited regimes partially recovered in 2023, their spatial
extent remained smaller than in 2021, likely due to ongoing $NO_x$ emission reductions outpacing changes
in VOCs emissions, contributing to a structural shift in ozone formation chemistry. These findings
highlight the influence of climate-induced VOCs responses and precursor imbalance in driving ozone
formation regime shifts and complicating ozone mitigation efforts. While this influence has already
become prominent in the current phase, it is expected to intensify with the increasing frequency of
extreme weather events in the future.

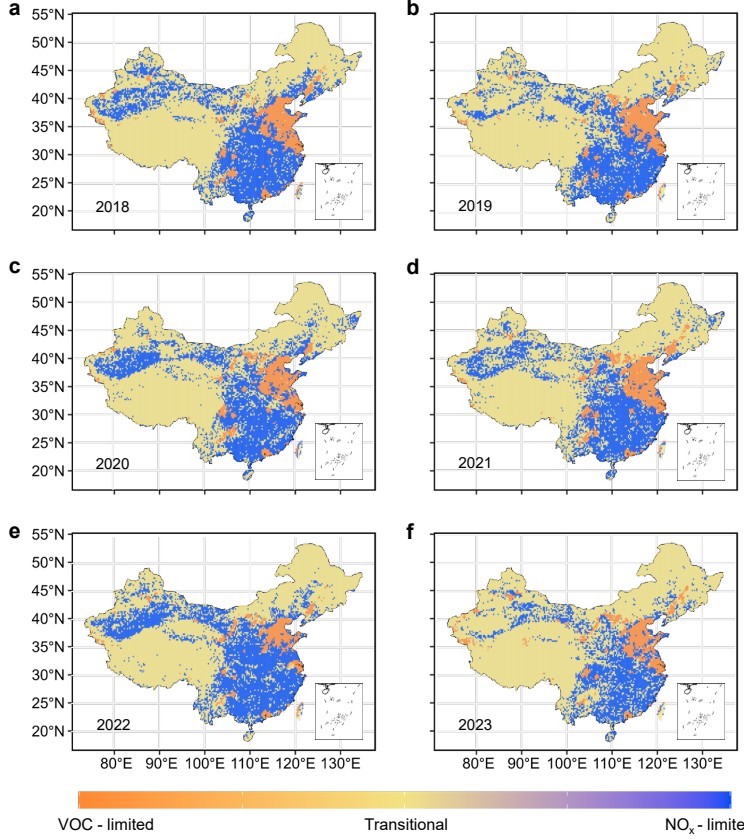


**Figure 3. Ozone formation sensitivity regimes.** The results of FNR analysis from June to August (2018-2023) are
presented, showing the spatiotemporal variation of ozone sensitivity in different regions. The colors in the map
represent the geographical distribution of VOC-limited, $NO_x$-limited, and transitional ozone sensitivity zones. The
city locations within the five key regions of China are shown in Fig. S10.

**3.4 Impact of meteorological variations on ozone**



Meteorological conditions directly and indirectly modulate surface ozone concentrations by
influencing photochemical reactions, vertical mixing, and dispersion processes (Li et al., 2019b; Li et al.,
2020). These effects exhibit strong regional and temporal heterogeneity across China. As shown in Fig.
2, during Phase I (2013–2017), meteorological contributions to summertime MDA8 ozone remained
modest, ranging from -4.8 to +3.9 µg m$^{-3}$. In Phase II (2017–2020), notable ozone reductions attributable
to meteorology were observed－ -14.4 ± 3 µg m$^{-3}$, -15.9 ± 3.8 µg m$^{-3}$, and -11.1 ± 2.4 µg m$^{-3}$ in FWP,
YRD, and SCB, respectively. These reductions accounted for 58 ± 12%, 77 ± 18%, and 80 ± 17% of the
total summertime MDA8 ozone reduction during this phase. however, these impacts remained smaller
than those from emission controls in BTH and PRD. In contrast, during 2020–2023, ozone trends became
increasingly influenced by meteorological anomalies, particularly in 2022. That summer, extreme
heatwaves (Mallapaty, 2022; Wang et al., 2023b) led to sharp increases in MDA8 ozone, contributing
20.8 ± 3.6 µg m$^{-3}$ in YRD and 22.1 ± 3.2 µg m$^{-3}$ in SCB. In 2023, however, abundant summer rainfall
suppressed ozone formation, with MDA8 ozone decreasing by –17.8 ± 2.3 µg m$^{-3}$ in YRD and –9.7 ± 3.3
µg m$^{-3}$ in SCB. These declines correspond to year-on-year increases in rainfall of 102% and 35% in the
two regions, respectively (Fig. S11).
To further elucidate the dominant meteorological drivers of ozone variability, we examined Gini
importance (Nembrini et al., 2018; Wright and Ziegler, 2017) scores derived from the RF model across
18 predictor variables (Fig. S12). Temperature ($T$) and relative humidity (RH) emerged as the most
influential variables in the BTH, FWP, and SCB regions, while in the YRD, shortwave solar radiation
(SR), RH, and rainfall were dominant. These results suggest that ozone variability is governed by
complex meteorological interactions that vary regionally. For instance, rainfall is typically associated
with lower solar irradiance and increased cloud cover, both of which are unfavorable for photochemical
ozone production (Jacob and Winner, 2009; Shan et al., 2008). Moreover, the high importance of $T$ and
SR in these regions indicates that surface ozone is highly sensitive to thermal conditions and
photochemical intensity (Yang et al., 2025). Elevated temperatures accelerate ozone precursor emissions
and reaction rates, while stronger solar radiation enhances photolysis and ozone formation potential (Qin
et al., 2025; Tao et al., 2024). In the PRD, ozone variability was more strongly influenced by temperature
and transport-related indices (such as meridional winds at different layers, etc.). This likely reflects the
region's subtropical coastal climate, where frequent summer typhoon incursions from the Northwest





Pacific modulate large-scale atmospheric transport (Chen et al., 2024; Wang et al., 2024a; Wang et al.,
2022b). These events may introduce strong horizontal advection and vertical mixing, thereby altering the
distribution and buildup of ozone precursors, and contributing significantly to the observed ozone
variability. As shown in Fig. S13, the correlations between ozone and key meteorological variables were
notably enhanced during heatwave (HW) periods. Specifically, ozone positively correlated with both $T$
and SR, and negatively (or weakly) correlated with RH. During prolonged rainfall (PR) events, cities in
the Yangtze-Huaihe region showed the strongest RH–ozone anti-correlation ($R < -0.7$), likely driven by
the enhanced wet scavenging and reduced photochemistry (Fig. S14 a – c).
To quantify the individual contributions of meteorological variables, we applied SHAP (SHapley
Additive exPlanations) analysis to HW and PR events in the Yangtze-Huaihe region from 2015 to 2023
(Supporting Methods S3). As shown in Fig. S15 and Fig. S14 d, HW events were associated with strong
positive SHAP values in southeastern coastal cities, the YRD, and SCB – primarily driven by elevated
SR and $T$. Indeed, mean SR during HW periods was significantly higher than during non-HW periods
(Fig. S16), amplifying photochemical ozone production potential. In contrast, PR events consistently
yielded negative SHAP contributions across all cities, mainly due to reduced sunlight and suppressed
precursor buildup. A multi-year comparison (Fig. 4) highlights the opposing effects of key meteorological
variables – including RH, $T$, boundary layer height, total precipitation, and surface pressure – on MDA8
ozone. SR, RH, and $T$ emerged as the most influential parameters, while total cloud cover and
meteorological transport playing secondary roles during HW episodes. The intensity of HW and PR
events modulated the magnitude of these effects. For instance, high-rainfall PR events in 2016 and 2020
yielded large negative SHAP contributions (–29.7 and –16.9 µg m$^{-3}$), mainly via RH-driven suppression.
Conversely, reduced rainfall in 2023 weakened the RH effect, though advection and vertical mixing still
contributed to ozone suppression (Fig. S17).

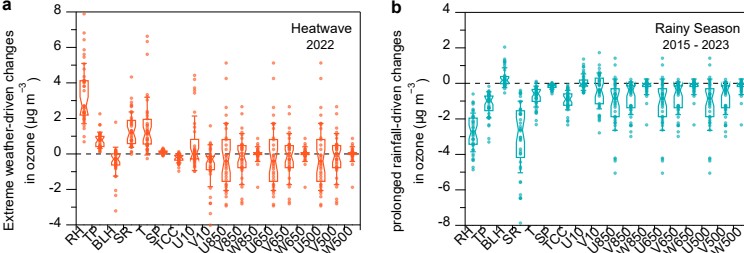


**Figure 4. Meteorological influences on predicted ozone concentrations under heatwave and rainy weather**
**conditions.** (**a**) Differences in SHAP values (ΔSHAP) between heatwave and non-heatwave periods in the Yangtze-





Huaihe region during summer 2022. (**b**) Differences in SHAP values (ΔSHAP) between prolonged rainfall periods
and non-prolonged rainfall periods in the same region from 2015 to 2023. Box plots show the distribution of ΔSHAP
across cities; the center line indicates the median, boxes denote the interquartile range (25th-75th percentiles), and
whisker line extends to one standard deviation.

**3.5 Reshaping ozone trends in a warming climate**

398        To assess the long-term influence of climate change on surface ozone concentrations, we applied

the FEA framework to simulate summertime ozone trends over the period 1970 – 2023. In this analysis,
anthropogenic emissions were held constant at their 2015 – 2023 summertime levels, while interannual
variations in meteorological variables were introduced using historical reanalysis data. This design
isolates the climate-driven component of ozone trends while assuming that emission trajectories are
independent of climate change – a simplification aligned with prior attribution frameworks (Wang et al.,
2022c). The impact of anthropogenic emission controls was estimated by comparing observed ozone
concentrations with FEA-predicted values during 2015 – 2023, thereby quantifying the residual effect of
emissions under fixed meteorology.

407        As shown in Fig. 5, under the 2015-2023 emission levels, climate change has exerted a statistically

significant ($p < 0.05$) positive influence on urban summertime ozone concentrations across China,
resulting in a nationwide increase of approximately 0.06 $\mu g\,m^{-3}\,a^{-1}$ since 1970. All five major urban
regions displayed upward trends, with the most pronounced increase observed in the BTH and SCB at
0.12 $\mu g\,m^{-3}\,a^{-1}$. Spatial correlations between climate-driven ozone increases and temperature changes
(Fig. S18) further confirm that warming is the dominant contributor to long-term ozone enhancement. In
particular, the correlation coefficients between ozone trends and temperature anomalies reached 0.90
(BTH), 0.89 (FWP), 0.72 (YRD), and 0.93 (SCB), indicating a strong temperature dependence of
climate-induced ozone formation in these regions. The PRD showed a weaker correlation, likely due to
its unique subtropical maritime climate and higher humidity and cloud cover, which tend to suppress
photochemical ozone production(Yang et al., 2019).

418        These findings are consistent with previous projections that forecast an increase in high-ozone

events under future climate scenarios spanning 2020–2100 (Li et al., 2023). The historical record already
reflects this risk: despite significant increases in anthropogenic emissions driving ozone growth prior to
2018, national air quality improvement measures began to yield reductions thereafter. However, since





2020, a rebound in ozone concentrations has emerged in several regions, suggesting that the climatic
penalty for ozone is beginning to offset the benefits of emission control. The intense heatwave of the
2022 summer significantly enhanced ozone formation and altered ozone sensitivity – shifting from a
VOC-limited region to a transition zone or $NO_x$-limited region. Meanwhile, the reduction in
anthropogenic emissions of ozone precursor substances indirectly led to changes in ozone sensitivity,
thereby making anthropogenic emission reductions more effective in ozone control. However, overall,
the direct effects of climate change (i.e., increased ozone formation) far outweigh the indirect effects of
anthropogenic emission controls, indicating that the punitive effects of climate change on ozone will
become increasingly significant in the future. Taken together, these results underscore the dual challenges
of air quality management in a warming climate. Anthropogenic emission reductions remain critical, but
they may no longer suffice in isolation. As the warming-driven enhancement of ozone formation becomes
more prominent, China and other rapidly urbanizing regions will require adaptive and climate-resilient
air quality strategies – including dynamic precursor control, land-use planning, and extreme weather
early warning systems –to sustainably mitigate ozone pollution in the decades to come.

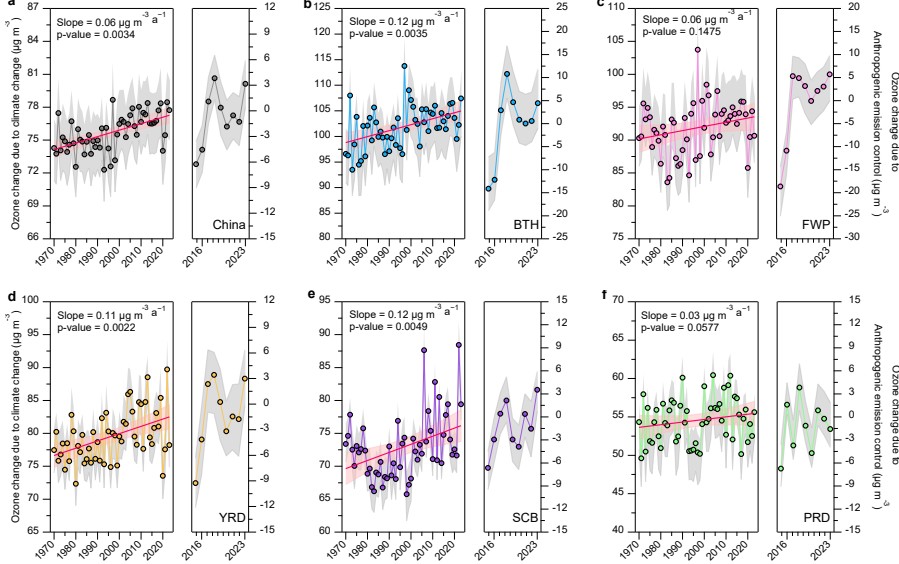


**Figure 5. Impact of climate change and emission controls on ozone trends.** Left panels show ozone trends
attributable to long-term climate change from 1970 to 2023, simulated under fixed emission scenarios using the FEA
framework. Right panels depict ozone trends from 2015 to 2023, reflecting the impact of anthropogenic emission
controls. Each trajectory represents results based on a distinct emissions baseline year. Shaded grey areas indicate
the interquartile range (25th-75th percentiles), solid red lines denote trend estimates, and light red shading marks the



5th-95th percentile confidence intervals. Statistical significance and trend slopes were assessed using the Mann-
Kendall test.

**4 Conclusions and implications**

China is confronted with the dual challenges of climate change and ozone pollution. Over the
past decade, summertime ozone concentrations across the country have exhibited complex
spatiotemporal patterns, reflecting the evolving interplay between anthropogenic emissions,
meteorological variability, and large-scale climate dynamics. In this study, we developed and
applied a machine learning-based FEA framework to disentangle and quantify the respective roles
of anthropogenic emissions and meteorological drivers in shaping ozone trends during 2013-2023.
With a national-level prediction uncertainty of approximately 6%, the FEA method provides a
computationally efficient and scalable tool for diagnosing atmospheric variability across large
spatial and temporal domains.
Our analysis revealed that increased anthropogenic precursor emissions were the dominant
driver of the sharp rise in summertime MDA8 ozone concentrations during the first phase (2013–
2017), contributing an average increase of $23.2 \pm 1.1$ µg m$^{-3}$. In contrast, during the second phase
(2018–2020), enhanced air quality regulations – particularly the synergistic control of NO$_x$ and
VOCs – led to measurable reductions in MDA8 ozone, with national-average declines of $4.6 \pm 1.5$
µg m$^{-3}$. These improvements were especially evident in regions such as BTH and FWP, where ozone
formation is highly sensitive to VOC levels. However, during the most recent period (2021–2023),
the impact of emission reductions diminished considerably, with regional ozone levels either
plateauing or rebounding. This stagnation underscores the urgent need for more targeted, region-
specific emission control strategies that address the shifting photochemical sensitivity of ozone
formation regimes.
Applying the SHAP method, we further quantified the impacts of extreme meteorological
events on ozone levels. Our results show that record-breaking heatwaves in 2022 contributed to
widespread ozone enhancements of up to 5.8 µg m$^{-3}$, while prolonged rainfall events – particularly
during the East Asian plum rain seasons – suppressed ozone production by as much as –15.2 µg m$^{-3}$.



These findings highlight the increasingly dominant role of short-term meteorological extremes in modulating ozone air quality under a warming climate. In parallel, satellite-based FNR analysis diagnostics revealed that most urban clusters in China remained in VOC-limited or transitional regimes, except the PRD, which was largely $NO_x$-limited. The 2022 heatwave triggered regime shifts in regions such as the YRD, where rising VOCs emissions and elevated temperatures shifted the photochemical regime toward $NO_x$-limited. These results emphasize the importance of dynamic, region-specific assessments of ozone formation sensitivity in the formulation of effective mitigation strategies.

To assess the climate penalty on ozone, we extended the FEA framework to simulate long-term trends from 1970 to 2023, by fixing emissions and allowing meteorological variables to evolve with observed climate trends. Our findings show that climate change has contributed a significant upward trend in urban summertime ozone, averaging 0.06 $\mu g\ m^{-3}\ a^{-1}$, with particularly strong increases in the BTH and SCB. Correlations between ozone and surface temperature were consistently high ($r = 0.72$–$0.93$) in BTH, FWP, YRD, and SCB, suggesting that warming has increasingly offset gains from emission controls in recent years.

While the FEA framework provides a powerful diagnostic tool, some limitations remain. For example, the historical simulations did not account for climate-driven changes in land use, topography, or population density, which may introduce biases in long-term attribution (Zhu et al., 2025). Future work could incorporate dynamic ancillary datasets and emissions scenarios to further improve model performance. Overall, this study underscores the escalating influence of climate extremes on ozone variability and the emerging limits of conventional emission control approaches. In the face of continued warming, machine learning-based attribution frameworks such as FEA offer a promising pathway for integrating meteorology, chemistry, and policy analysis. To achieve sustained improvements in ozone air quality, future strategies must consider the compound effects of anthropogenic emissions, short-term weather events, and long-term climate change, and adopt adaptive, region-specific, and climate-resilient air quality management frameworks.



*Data availability.* Data are provided within the manuscript or supplementary information files.

*Code availability.* The statistical computing in this study was based on R language software which can be download at https://www.r-project.org/.

*Author contributions.*

Y.Z. and X.G. initiated and designed the study. Y.Z. and JF developed the statistical methodology, model calculation, and data analysis. J.F. and Y.Z. prepared the manuscript with contributions from D.H., B.Z., M.W., J.L., Y.S., H.L., J.W., Y.W., M.C., and X.G..

*Competing interests.* The authors declare no competing interests.

*Acknowledgments.*

This study was supported by the National Natural Science Foundation of China (grant no. 42207124) and Natural Science Foundation of Jiangsu Province (grant no. BK20210663).

**Correspondence** and requests for materials should be addressed to Yunjiang Zhang or Xinlei Ge.

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
