# Peer review of "Tracking surface ozone responses to clean air actions under"

_EGUsphere, 2025_

## Author Comment (AC1)

**Response to referee #2**

Comments: Tropospheric ozone is a globally important air pollutant and a short-lived climate forcer, with substantial impacts on human health, climate change, and terrestrial ecosystems. Understanding the relationship between ozone concentration changes and their driving factors is essential for developing effective control strategies. This study utilizes ground-based observational data from the Chinese monitoring network during 2013-2023 and develops a machine-learning-based method to quantitatively disentangle the contributions of meteorological conditions and anthropogenic emissions. The analysis is further extended to evaluate the sensitivity of ozone to climate change. In addition, the authors employ satellite retrievals to explore the changes in precursor ratios and to diagnose the shifts in chemical regimes. The proposed analytical framework provides valuable insights and will be highly informative for future studies. Overall, the manuscript is clearly structured, well-designed, and well-written, and it fits well within the scope of ACP. I would recommend publication after the following issues are addressed:

**Response:** We sincerely thank the reviewer for the encouraging and insightful evaluation of our study. We are very grateful for the recognition of our analytical framework and its relevance to understanding ozone evolution and control strategies. Following the reviewer's valuable suggestions, we have carefully revised the manuscript to improve its clarity and completeness. All corresponding modifications have been incorporated into the revised version, with detailed point-by-point responses provided below.

Title suggestion: Consider revising the title to "Tracking surface ozone responses to clean air actions under a warming climate in China" for clarity and stronger alignment with the scope.

**Response:** We greatly appreciate the reviewer's constructive suggestion regarding the title. Following this advice, we have revised the title to:

"Tracking surface ozone responses to clean air actions under a warming climate in China using machine learning."

This revised title provides clearer expression and more accurately captures the methodological framework and research scope of our study.

Lines 52-54: This sentence requires additional references. In particular, the 2021 IPCC report should be cited and carefully verified.

**Response:** We appreciate the reviewer's helpful suggestion. In response, we have thoroughly reviewed the relevant literature and added the IPCC Sixth Assessment Report (AR6, 2021) as an authoritative reference to strengthen and substantiate the statement in lines 57–58. The corresponding

text has also been carefully checked and revised to ensure both scientific accuracy and contextual consistency.

Line 81: Provide the full name of XGBoost when first introduced.

**Response:** Thank you for your suggestion. We have now provided the full name of XGBoost ("eXtreme Gradient Boosting") upon its first mention in line 85-86 to improve clarity and readability.

Line 89: Please remove the word "monsoon".

**Response:** Thank you for your suggestion. This word "monsoon" has been removed in the revised manuscript.

Line 109: The study develops an innovative machine-learning framework for attribution analysis, including an extension to climate change. This is a key contribution, but I suggest adding more technical details, such as a conceptual diagram of the methodology, to improve clarity and accessibility for readers.

**Response:** We sincerely appreciate the reviewer's positive recognition of our machine-learning-based analytical framework. To improve the clarity and accessibility of the methodology, we have added a conceptual diagram (Figure 1) that outlines the overall workflow of the framework and its application to trend attribution and climate change impact assessment. This addition visually summarizes the key analytical steps and enhances the reader's understanding of the underlying processes and logical structure of the study.

[Figure]

***Figure 1. Schematic framework of data analysis and methodology.*** *This study integrates multi-dimensional datasets, including ground-based observations, meteorological reanalysis, and satellite remote sensing. A fixed emission approximation (FEA) approach, developed based on the random forest (RF) model, is employed to quantitatively disentangle the contributions of meteorological conditions (MET) and anthropogenic emissions (ANT) to ozone trend variations, and its performance is compared with the conventional meteorological normalization method. The SHAP technique is further applied to assess the influence of extreme weather events, such as heatwaves (HW) and extreme precipitation (PR). The satellite-derived formaldehyde-to-nitrogen dioxide ratio (FNR) is used to diagnose ozone production sensitivity, to explain and verify the impact of extreme weather and anthropogenic emissions on ozone. Finally, the FEA framework is extended to evaluate the long-term impacts of climate change on ozone trends since 1970.*

Line 114: The role of time variables requires clarification. Were the diurnal and seasonal/monthly variables included to remove short-term and seasonal variability, leaving the long-term trend for quantitative attribution? Please explain explicitly.

**Response:** We thank the reviewer for this insightful comment. Yes, the diurnal and seasonal/monthly variables were incorporated as proxies for short-term, periodic variations in emissions and meteorological conditions. Their inclusion allows the model to effectively separate these regular

temporal patterns from the long-term interannual trends that are the main focus of our quantitative attribution analysis. The revised manuscript now clarifies this point, and the relevant text has been updated to read as follows:

*The time variables – hour (hour of day) and month (month of year) – are used as emission surrogates to capture regular diurnal and seasonal variations in anthropogenic activity. A similar strategy is widely applied in previous studies about long-term trends in air pollutants (e.g., Grange et al., 2018; Vu et al., 2019) to separate short-term cyclical emission variability from long-term trends.*

*These temporal emission surrogates, including month and hour, represent short-term regular emission patterns (e.g., diurnal cycles), thereby enabling the model to isolate the long-term emission-driven component of ozone changes (Grange et al., 2018; Meng et al., 2025; Shi et al., 2021; Vu et al., 2019).*

Line 125: Why was the modeling performed separately for each city, rather than by grouping cities into regions? Please explain the rationale.

**Response:** Thank you for this valuable comment. We conducted the modeling separately for each city to minimize uncertainties arising from surface and emission heterogeneity within broader regions. Cities across China exhibit distinct characteristics in terms of land use patterns, emission structures, local meteorology, and boundary-layer dynamics, all of which can strongly influence ozone formation and variability. Modeling at the city level allows the framework to better capture these localized processes and maintain higher fidelity in attribution analysis. We have revised the manuscript to clarify this rationale, as follows:

*Our modeling strategy involves building and predicting models for individual cities and for each year from 2015 to 2023, which helps in minimizing the uncertainty caused by surface heterogeneity.*

Nonetheless, we acknowledge that this approach may also introduce certain limitations. Specifically, the current implementation does not explicitly resolve grid-scale spatial heterogeneity, vegetation activity, or land-use dynamics, which may influence local ozone formation. To address this, we have included additional discussion in the conclusion section as follows:

*Nonetheless, some limitations remain. The current implementation did not explicitly resolve grid-scale spatial heterogeneity, vegetation, or land-use dynamics, which may influence ozone formation. Moreover, potential sensitivities to spatial resolution warrant further investigation through coupled applications of machine learning and chemical transport models.*

Lines 152-161: The uncertainty analysis, particularly for the Fixed Emission Approximation (FEA)

method, is highly valuable. I strongly recommend moving these results (currently Figure S3) into the main text.

**Response:** We fully agree with the reviewer that the uncertainty analysis of the Fixed Emission Approximation (FEA) method represents a key component of the study. To improve its visibility and enhance the transparency of our methodological evaluation, we have moved these results from the Supplementary Information (previously Figure S3) to the main text as the new Figure 2. This adjustment allows readers to more directly assess the robustness and reliability of the FEA framework, thereby strengthening the methodological clarity and scientific rigor of the paper.

Lines 202-205: The manuscript highlights several regions in China. Please explain why these regions were emphasized and include a map showing their geographic distribution for better context.

**Response:** We thank the reviewer for this valuable comment. The selected regions – BTH, YRD, FWP, SCB, and PRD – were chosen because they are representative urban clusters of different parts of China and capture the diversity in emission characteristics and atmospheric conditions. For example, the BTH region represents northern inland cities dominated by anthropogenic emissions, while the PRD region represents southern coastal cities with substantial biogenic emissions. The YRD exhibits strong anthropogenic emissions influenced by southern biogenic sources, and SCB reflects the pollution characteristics of central and southwestern China. These regional divisions are consistent with prior studies and provide a meaningful framework for analyzing ozone variability across China. To enhance clarity, we have added a new map illustrating the geographic distribution of these regions, now included as Figure 3 in the revised manuscript. This addition allows readers to easily contextualize the regional analyses. The relevant text has been updated to read as follows:

*Figure 3 presents the spatial distribution of the average summertime (2018-2023) maximum daily 8-hour average (MDA8) ozone, surface $NO_2$, and TROPOMI $NO_2$, HCHO column concentrations across China, along with the locations of the country's five major city clusters: Beijing-Tianjin-Hebei (BTH), Fenwei Plain (FWP), Yangtze River Delta (YRD), Sichuan Basin (SCB), and Pearl River Delta (PRD). Across these five major city clusters, the average summer ozone concentrations ranged from 88.9 to 161.3 $\mu g\ m^{-3}$ – substantially exceeding the 43.0 $\mu g\ m^{-3}$ threshold associated with ecosystem productivity loss (Gong et al., 2021) and the World Health Organization (WHO, 2021)-recommended peak seasonal average of 60 $\mu g\ m^{-3}$. TROPOMI satellite observations of $NO_2$ column concentration show notably elevated concentrations over the five major city clusters, particularly in the BTH, YRD, and FWP, which align with surface $NO_2$ distribution patterns and*

*confirm the scale of anthropogenic NO$_x$ emissions in these regions (Zheng et al.,2021). TROPOMI satellite observations of HCHO column concentrations similarly reveal these city clusters as hotspots for VOC emissions (Fig. 3d). These concurrent high levels of NO$_2$ and HCHO suggest a strong photochemical ozone pollution potential, as the abundant precursors in these urban clusters could drive substantial ozone production during the summer months. This highlights the significant risks posed by summertime ozone in China's most urbanized and industrialized regions, with implications for both human and ecosystem health.*

[Figure]

**Figure 3. Spatial distribution of summertime MDA8 ozone, surface NO$_2$, and TROPOMI NO$_2$, HCHO across major city clusters in China.** *The panels represent the average MDA8 ozone, surface NO$_2$, and TROPOMI NO$_2$, HCHO column concentrations for 354 cities in China during the summertime (June–August) from 2018 to 2023. The corresponding five regions includes BTH (37°–41°N, 114°–118°E); YRD (30°–33°N, 118.2°–122°E); SCB (28.5°–31.5°N, 103.5°–107°E); PRD (21.5°–24°N, 112°–115.5°E) and FWP (106.25–111.25°E, 33–35°N, and 108.75–113.75°E, 35–37°N).*

Line 227: The references here primarily address ecological impacts, yet the text mentions "both human and ecological health." Please provide more references specific to human health. Also, revise "ecological health" to "ecosystem health."

**Response:** We thank the reviewer for this helpful suggestion. In response, we have added references specifically addressing the impacts of tropospheric ozone on human health, including respiratory and cardiovascular outcomes. Additionally, we have revised the terminology from "ecological health" to "ecosystem health" throughout the manuscript to ensure precision. The modified text now reads:

*Across these five major city clusters, the average summer ozone concentrations ranged from 88.9 to 161.3 μg m$^{-3}$ – substantially exceeding the 43.0 μg m$^{-3}$ threshold associated with ecosystem productivity loss (Gong et al., 2021) and the World Health Organization (WHO, 2021)-recommended peak seasonal average of 60 μg m$^{-3}$.*

*This highlights the significant risks posed by summertime ozone in China's most urbanized and industrialized regions, with implications for both human and ecosystem health.*

Line 229: The phrase "reflecting initial policy effectiveness" is unclear. Please rephrase for precision.

**Response:** We thank the reviewer for pointing this out. To avoid potential confusion, we have removed the phrase entirely, as Fig. S1 focuses solely on observed concentration changes rather than explicitly quantifying policy impacts. This modification enhances clarity and precision in the text.

Lines 230-232: The conclusion drawn here seems overstated, as the evidence provided is insufficient. This section mainly discusses temporal and spatial ozone concentration trends. A more cautious interpretation is recommended: instead of attributing trends directly to policy effectiveness, the authors could note that observed trends occurred under varying emission control backgrounds, while meteorology also played an important role.

Response: We appreciate the reviewer's suggestion. The text has been revised to adopt a more cautious interpretation, emphasizing that the observed temporal and spatial ozone trends occurred under varying emission control contexts and were influenced by meteorological variability. The revised sentence now reads:

*Spatially, ozone hotspot regions expanded between 2013 and 2017 (Fig. S1 a-e), followed by contraction during 2018-2020 (Fig. S1 f-i). However, this progress stalled in 2021. A sharp reversal was observed in 2022, with widespread increases in MDA8 ozone (Fig. S1 k). These changes could be closely linked to emission control measures and meteorological conditions, which will be further discussed in Sections 3.2 and 3.3.*

Line 243: Please define the parameters τ and p shown in Figure 1.

**Response:** We thank the reviewer for this suggestion. τ is a statistic used in the Mann-Kendall trend test to measure the correlation between data points in a sequence, but it is rarely used. The p-value is a statistic used to assess the statistical significance of the trend. We have now removed τ and explicitly defined the parameter p in the figure caption of Figure.

Line 276: Correct "Emission-driven" to "emission-driven."

**Response:** Thank you for pointing this out. We have corrected the capitalization, changing "Emission-driven" to "emission-driven" in the revised manuscript.

Lines 279-281: The logic here is confusing. The discussion first emphasizes the role of anthropogenic emissions, but then suggests that changes in emissions highlight the role of meteorology. Please clarify or restructure this argument.

**Response:** Thank you for your comment, and sorry for the confusion. We have restructured the argument for clarity. The revised text now reads:

*These results indicate that while emission control policies initially produced substantial benefits, their effectiveness has gradually diminished, suggesting that ozone responses to further emission reductions may have reached a saturation point.*

Lines 299-300: Please rephrase the sentence. The term "near-baseline" is ambiguous and requires clarification.

**Response:** Thank you for pointing this out. We have rephrased the sentence for clarity:

*By 2023, HCHO concentrations had returned to pre-heatwave levels.*

Line 342: There is an editorial error that needs correction.

**Response:** Thank you for your comment. We have identified the editorial error and corrected it. The revised text now reads:

*Ozone decreases attributable to meteorology reached $-14.4 \pm 3.0$ µg m$^{-3}$ in the FWP, $-15.9 \pm 3.8$ µg m$^{-3}$ in the YRD, and $-11.1 \pm 2.4$ µg m$^{-3}$ in the SCB, explaining $58 \pm 12\%$, $77 \pm 18\%$, and $80 \pm 17\%$ of the total ozone decline, respectively.*

Line 390: I recommend revising the y-axis labels for greater accuracy. For instance, in panel (a), the label currently suggests "extreme weather," but it actually represents only "extreme heatwave". In contrast, panel (b) provides a more specific description. The labeling should be made consistent and precise to avoid potential misinterpretation.

**Response:** Thank you for pointing this out. We have revised the y-axis labels to improve clarity and consistency. Specifically, panel (a) now explicitly indicates "Extreme Heatwave (HW)," while panel (b) retains the more specific description of pluvial events. This ensures accurate representation and avoids potential misinterpretation. The updated figure has been included in the revised manuscript.

[Figure]

*Figure 7. Meteorological impact on predicted ozone concentrations under heatwave and rainy weather conditions. (a) Differences in SHAP values (ΔSHAP) between heatwave and non-heatwave periods in the Yangtze-Huaihe region during summer 2022. (b) Differences in SHAP values (ΔSHAP) between prolonged rainfall periods and non-prolonged rainfall periods in the same region from 2015 to 2023. Box plots show the distribution of ΔSHAP across cities; the center line indicates the median, boxes denote the interquartile range (25th-75th percentiles), and whisker line extends to one standard deviation.*

The title of this section should be revised, since the authors are not reconstructing the ozone trend per se. A more accurate option could be "Reshaping distributions of ozone controlled by a warming climate." This section is indeed interesting and methodologically innovative. However, the manuscript should elaborate more clearly on which specific factors are included in the climate-change-driven trend, especially considering the constraints posed by the limited length and coverage of historical observational records.

**Response:** Thank you for your insightful comments and suggestions. We have revised the title to "Reshaping distributions of ozone controlled by a warming climate" to more accurately reflect the content. Additionally, we have clarified the factors included in the climate-change-driven trend. Specifically, the trend incorporates temperature increases. We also note the constraints imposed by the limited length and spatial coverage of historical observational records, which are acknowledged in the discussion.

Line 331 (Figure 3): Ensure the map format is consistent with that in the Supplementary Figures.

Response: Thank you for your comment. In the revised manuscript, we have replaced the corresponding Figure with a map showing the trend of ozone sensitivity intervals across the five major city clusters of China from 2018 to 2023. Additionally, the map in the supplementary material has also been updated to reflect the latitude and longitude range corresponding to the five major city

clusters. **The related modification is shown as Fig. 6 and Fig. S11.**

Lines 374-376: Since the SHAP interpreter is a key tool used to analyze predictor contributions, it should be briefly described in the Methods section.

**Response:** Thank you for your comment. We have moved the relevant description of the SHAP interpreter from the supplementary material to the main body of the manuscript. The updated content can now be found in Section 2.4.

Line 437: In Figure 5, the authors present both climate-change-driven and emission-driven trends. I am curious about how the results from the proposed FEA method compare with those from other widely used machine-learning approaches for trend analysis, such as de-weather. A comparison between different methods would not only be interesting but also serve as a useful validation of the robustness of the proposed framework.

**Response:** We thank the reviewer for this valuable suggestion. To address this, we have incorporated an analysis using the widely used "weather normalization" method and compared the results with those obtained from our FEA framework. The comparison shows that the trends derived from both approaches are highly consistent, demonstrating the robustness and reliability of our proposed FEA methodology. The inter-comparison results have been added to the revised manuscript for transparency and validation. The relevant text has been updated to read as follows:

*2.3 Weather normalization analysis*

*To compare the FEA method with other commonly used statistical approaches, we also applied the widely adopted meteorological normalization technique based on the RF algorithm. This approach constructs a regression model that relates air pollutant concentrations to meteorological parameters and emission surrogate indicators (i.e., time variables such as unix time, day of year, day of month, and hour of day) (Grange et al., 2018; Vu et al., 2019). Once the model is trained, pollutant concentrations are predicted by randomly resampling meteorological variables from long-term historical meteorological datasets, thereby generating a new ensemble of predictions (Vu et al., 2019). These predictions are made under consistent meteorological conditions, enabling the isolation of meteorological influences from anthropogenic emission effects on air pollutant trends. The resulting weather-normalized pollutant concentrations (Fig. 1) represent the levels expected under average meteorological conditions, thus reflecting the impact of emission changes alone. This approach, first proposed by Grange et al. (2018), has been widely applied in the long-term attribution of air pollution trends and in assessing short-term emission reduction effects (Shi et al., 2021; Vu et al., 2019). In this study, the meteorological normalization follows this established framework, with meteorological*

*variables randomly sampled from the long-term dataset spanning 1970-2023. Each normalization process involves 1,000 iterations, and the arithmetic mean of these iterations' simulated values is adopted as the final normalized result. The alignment between FEA-based and weather-normalized trends (Fig. S4) affirms the robustness of the FEA framework.*

*Supplement:*

[Figure]

***Figure S4. Trends in the average summertime ozone concentration changes from 2015 to 2023, driven by anthropogenic emission control.*** *The figure compares the ozone trend variations for six representative cities in key regions, based on both the FEA method and weather normalization method.*

Line 445 (Conclusion): The conclusion is overly lengthy. Please condense and refine this section for clarity and impact.

**Response:** Thank you for your suggestion. We have revised the conclusion to make it more concise and impactful, emphasizing the key findings and implications of our study.

---

## Author Comment (AC2)

**Response to Referee #1**

General Comments: The manuscript provides valuable insights into surface ozone trends in China, driven by emissions and meteorological factors, using the Fixed Emission Approximation (FEA) method. While the study is important, several key issues require attention before it can be accepted.

**Response:** We sincerely thank the reviewer for the positive and constructive comments on our manuscript. These suggestions have been very helpful in improving the quality and clarity of our work. We have carefully revised the manuscript accordingly.

1. The explanation of emission surrogates and the impact of coarse-resolution meteorological data on the regression process is insufficient. Additionally, regional ozone trends, especially in the PRD, need clearer explanations, and the manuscript should include maps or time-series for key ozone precursors (CO, $NO_x$, VOCs).

**Response:** Thank you for these important and constructive comments. We have addressed each point in the revised manuscript; a concise summary of our changes and rationale follows.

**(1)** We expanded the method section to clarify that the time variables (hour of day, month of year) serve as emission surrogates that capture regular diurnal and seasonal patterns in anthropogenic activity (e.g., traffic emission cycles). This approach is widely used in weather-normalization and RF-based attribution studies because it helps separate short-term cyclical emission variability from long-term trends (such as Grange et al., Vu et al., and Shi et al.). We also note that inclusion of these surrogates improved model performance in cross-validation. The detailed modification can be found in the revised manuscript on page 7 lines 140-143 and page 9 lines 197-198.

**(2)** We now explicitly state our handling of meteorological inputs: meteorological predictors were taken from the nearest ERA5 grid cell (0.25°×0.25°) to each city region, while the surface air pollutant concentrations represent multi-site city averages (i.e., averages over all available monitoring stations within each city). This city-average / nearest-grid strategy follows common practice in recent machine-learning air-quality studies and balances spatial representativeness with data availability. We have added the discussion of the limitations introduced by using coarse reanalysis fields and how this uncertainty was partially mitigated by (i) training models per city, (ii) using many meteorological predictors at multiple levels, and (iii) performing interannual resampling uncertainty tests (see new text and Fig. 2 now highlighted in the main text). We also discuss for future coupled machine learning and chemical transport modeling work to more fully assess resolution sensitivities. The detailed modification can be found in the revised manuscript on page 7 lines 144-149 and pages 22-23 lines 521-525.

**(3)** We expanded the results and discussion to better explain why PRD trends differ from northern regions. Key points added: (i)PRD has relatively stronger biogenic VOC and marine influences and lower anthropogenic $NO_x$ compared with northern basins (e.g., BTH), leading to a more $NO_x$-limited photochemical regime; (ii) subtropical maritime climate, higher humidity and cloudiness, and frequent typhoon/monsoon perturbations modulate transport and photolysis, reducing the straightforward response of ozone to local emission changes; (iii) therefore, PRD shows smaller ozone increases in response to the same emission changes that produced larger effects in VOC-sensitive northern regions. During phase II, the trends obtained using the FEA method were generally consistent with those from previous studies using statistical methods. In this phase, the changes in ozone production sensitivity regimes in the PRD region were more pronounced compared to those in the YRD and SCB regions. We cite recent regional studies that support these mechanisms. The detailed modification can be found in the revised manuscript on page15 line 340-342 and page17 lines 387-390.

**(4)** We agree that precursor maps/timeseries strengthen the attribution. Accordingly, we added: (i) spatial maps of surface CO and $NO_2$ (ground-based where available) and column $NO_2$ and HCHO from TROPOMI (new Fig. 3); (ii) city-level summertime and whole year time series for CO, $NO_2$, and $PM_{2.5}$ for the five major regions (new Fig. S5); and (iii) discussion linking these precursor trends to regime shifts ($HCHO/NO_2$) and the FEA results (Fig. 6). Because continuous ground-based VOC observations are sparse nationally, we did not add nationwide VOC maps, instead we use satellite HCHO as an established proxy for VOC emissions. The detailed modification can be found in the revised manuscript on page12 lines 277-292, page15 lines 342-343 and page15 lines 355-356.

[Figure]

**Figure 3. Spatial distribution of summertime MDA8 ozone, surface NO₂, and TROPOMI NO₂, HCHO across major city clusters in China.** The panels represent the average MDA8 ozone, surface NO₂, and TROPOMI NO₂, HCHO column concentrations for 354 cities in China during the summertime (June–August) from 2018 to 2023. The corresponding five regions includes BTH (37°–41°N, 114°–118°E); YRD (30°–33°N, 118.2°–122°E); SCB (28.5°–31.5°N, 103.5°–107°E); PRD (21.5°–24°N, 112°–115.5°E) and FWP (106.25–111.25°E, 33–35°N, and 108.75–113.75°E, 35–37°N).

[Figure]

**Figure S5. Ground-based observed time series of NO₂, CO and PM₂.₅.** Panels (a-c) show the summertime average time series of NO₂, CO, and PM₂.₅ for China's five major city clusters from 2015 to 2023. Panels (e-f) present the annual average time series of NO₂, CO, and PM₂.₅ for China's five major city clusters from 2015 to 2023.

2. The discussion on the COVID-19 lockdown requires more detailed analysis of TROPOMI data to explain regional ozone changes, and maps for regions like the North China Plain would help clarify spatial patterns. Figures, particularly Figure 3, lack clear quantitative results, and Section 3.3 requires deeper analysis.

**Response:** We sincerely thank the reviewer for these valuable and constructive comments. We have substantially revised the relevant sections to strengthen the discussion and quantitative analysis related to the COVID-19 lockdown period, as summarized below:

(1) We applied the formaldehyde-to-NO₂ ratio (FNR) diagnostic to classify ozone formation regimes and evaluate their spatial changes during the lockdown. The new analysis reveals clear regional differences: the North China Plain (NCP) and Yangtze River Delta (YRD) shifted toward more VOC-limited conditions due to sharp NOₓ reductions, while parts of southern China remained in NOₓ-limited or transition regimes. The updated manuscript includes corresponding maps to illustrate these spatial patterns (Fig. S13).

(2) We have significantly revised Figure 6 to provide year-round quantitative results, distinguishing the relative contributions of the three ozone sensitivity regimes (NOₓ-limited, VOC-limited, and transition) over time.

[Figure]

**Figure S13. Ozone formation sensitivity regimes during COVID-19.** Spatial distribution of ozone formation sensitivity regimes in China from January to April 2020 during the COVID-19 pandemic. The hollow triangles represent the geographical coordinates of five key urban clusters in China.

[Figure]

**Figure 6. Trends in the distributions of ozone production sensitivity regimes.** Fractions of VOC-limited, NOx-limited, and transitional ozone sensitivity regimes across five key regions during the summertime (June to August) from 2018 to 2023, based on the FNR analysis. Panel (f) presents the overall trends for all five regions.

3. Finally, the roles of temperature, humidity, and radiation in ozone formation need clearer interpretation.

**Response:** We appreciate this important and constructive comment. In response, we have expanded our analysis and discussion to clarify the distinct roles of temperature, humidity, and solar radiation in ozone formation across different regions. Our extended machine learning analysis (see Fig. S16) reveals region-specific sensitivities of ozone to these meteorological factors. We also incorporated relevant literature to further support the interpretation of these regional contrasts. The detailed

modification can be found in the revised manuscript on pages 18-19 lines 430-442.

[Figure]

**Figure S16. Partial dependence of ozone on T, RH, and SR for representative cities.** Panels show the 3D-dependence plots of MDA8 ozone with RH, T, and SR for representative cities in BTH, FWP, YRD, and PRD, including MDA8 ozone-RH-T, MDA8 ozone-RH-SR, and MDA8 ozone-SR-T.

In conclusion, a major revision is needed to address these issues, strengthen the analysis, and improve the clarity and presentation of the results.

**Response:** We sincerely thank the reviewer for the constructive and insightful comments. We have

carefully revised the manuscript to address all the concerns raised and substantially improved the analysis, clarity, and presentation of the results. Detailed point-by-point responses and corresponding revisions are provided below.

Specific Comments:

Line 193: What is TAP?

**Response:** Thank you for your comment. We have clarified the meaning of TAP in the revised manuscript:

*For 2013 – 2014, the surface MDA8 ozone data were obtained from the Tracking Air Pollution in China (TAP) dataset (Geng et al., 2021), which can be downloaded from http://tapdata.org (last accessed: May 20, 2024).*

Line 114: How does the author account for the uncertainty in coarse-resolution meteorological variables and site data during the RF regression process? Is the specified elevation referring to the site elevation or the average elevation of the coarse-resolution meteorological variable grid?

**Response:** Thank you for your comment. We now explicitly state our handling of meteorological inputs: meteorological predictors were taken from the nearest ERA5 grid cell (0.25°×0.25°) to each city region, while the surface air pollutant concentrations represent multi-site city averages (i.e., averages over all available monitoring stations within each city). This city-average / nearest-grid strategy follows common practice in recent machine-learning air-quality studies and balances spatial representativeness with data availability. We compared the observed meteorological data from weather stations in Nanjing, Suzhou, and Xuzhou with the ERA5 data we used and tested the sensitivity of ozone to different meteorological datasets. We found that the parameters from ERA5 closely aligned with those from the weather stations. Sensitivity tests using different meteorological datasets showed that the predicted ozone trends under fixed 2019 emission conditions were generally consistent. This supports the reliability of ERA5 data, which, compared to weather station observations, provides a more comprehensive dataset, therefore, we chose ERA5 data for our machine learning simulations. The following compares the hourly temperature, pressure, and relative humidity observed at weather stations in Nanjing, Suzhou, and Xuzhou in 2019 with the corresponding ERA5 data for these three parameters. Additionally, we replaced these three parameters with the observed data during model training and assessed the sensitivity of ozone to different meteorological datasets:

[Figure]

[Figure]

Additionally, we have added a discussion in the main text regarding the limitations introduced by using coarse reanalysis fields and how this uncertainty was partially mitigated by (i) training models for each city, (ii) using a wide range of meteorological predictors at multiple levels, and (iii) performing interannual resampling uncertainty tests. We also discuss the potential for future work that couples machine learning and chemical transport modeling to more fully assess the sensitivities to spatial resolution. The revised text now reads as follows:

*It should be noteworthy that surface air pollutant observations for each city represent multi-site averages rather than data from a single monitoring station, which reduces the influence of local representativeness errors. The meteorological data are obtained from the nearest grid cell corresponding to each city, ensuring spatial consistency between the pollutant and meteorological datasets. This approach was similar to the methodologies widely adopted in previous studies (Shi et al., 2021; Wang et al., 2025; Yao et al., 2024; Zheng et al., 2023). Our modeling strategy involves building and predicting models for individual cities and for each year from 2015 to 2023, which helps in minimizing the uncertainty caused by surface heterogeneity.*

*Nonetheless, some limitations remain. The current implementation did not explicitly resolve*

*grid-scale spatial heterogeneity, vegetation, or land-use dynamics, which may influence ozone formation. Moreover, potential sensitivities to spatial resolution warrant further investigation through coupled applications of machine learning and chemical transport models.*

Line 115: What is an emission surrogate? Where did the author obtain it? It should be mentioned in Section 2.1.

**Response:** Thank you for this helpful comment. We have expanded Section 2.1 to clarify the meaning and source of the emission surrogates. Specifically, the time variables – hour of day and month of year – are used as emission surrogates to capture regular diurnal and seasonal variations in anthropogenic activity (e.g., traffic emissions). This approach is widely applied in weather-normalization and RF-based attribution studies (e.g., Grange et al., 2018; Vu et al., 2019) to separate short-term cyclical emission variability from long-term trends. We also found that including these surrogates improved model performance during cross-validation. The revised text now reads as follows:

*The time variables – hour (hour of day) and month (month of year) – are used as emission surrogates to capture regular diurnal and seasonal variations in anthropogenic activity. A similar strategy is widely applied in previous studies about long-term trends in air pollutants (e.g., Grange et al., 2018; Vu et al., 2019) to separate short-term cyclical emission variability from long-term trends.*

*To assess uncertainty stemming from interannual model training variability, we applied a matrix-based resampling approach (see Text S2). As shown in Fig. 2, the relative difference in residuals ranged from -9% to 3%, and remained within ±12% for all regions – supporting the robustness of the FEA method. We found that the model with the added time variables exhibited significantly smaller uncertainty compared to the model without it (Fig. S3).*

[Figure]

***Figure 2. Uncertainty assessment of the FEA method.*** *The uncertainty for the FEA method is calculated using the approach described in Text S2. The diagonal line in each sub-panel represents the changes in the residuals of the models.*

[Figure]

***Figure S3. Uncertainty of the FEA method without time variables.*** *The uncertainty for the FEA method is calculated using the approach described in Text S2. The diagonal line in each sub-panel represents the changes in the residuals of the models.*

Line 117: "The aforementioned variables": It's not clear.

**Response:** Thank you for your comment. We have clarified the reference to "the aforementioned variables" in the revised manuscript:

*First, a regression model is constructed using the random forest (RF) algorithm to relate ozone concentrations to temporal emission surrogates and to meteorological parameters at multiple atmospheric levels. These temporal emission surrogates, including month and hour, represent short-term regular emission patterns (e.g., diurnal cycles), thereby enabling the model to isolate the long-term emission-driven component of ozone changes (Grange et al., 2018; Meng et al., 2025; Shi et al., 2021; Vu et al., 2019). The meteorological parameters include 18 distinct variables at different altitudes (see Table S1).*

Line 127-129: Rewrite this sentence.

**Response:** Thank you for your comment. We have rewritten the sentence for clarity:

*Specifically, we establish hourly-resolution models for the baseline year (i) during the summer*

*season (June to August) as a reference for anthropogenic emissions, represented by the pink solid line in Fig. 1.*

Line 243: "Anthropogenic drivers" to "Anthropogenic emission drivers".

**Response:** Thank you for your suggestion. We have updated it.

Line 243: You should show the anthropogenic emission map or time-series line for major ozone precursor such as CO, $NO_x$, and VOC.

**Response:** Thank you for your suggestion. We have added Ground-based observed time series of $NO_2$, CO and $PM_{2.5}$. The related modification is shown as follows:

*As shown in Fig. S5, the precursor gases $NO_2$ and CO exhibited regionally distinct decreasing trends, partially explaining the spatial heterogeneity of ozone changes.*

*In the post-pandemic period (2020–2023), concentrations of $NO_2$, CO, and $PM_{2.5}$ stabilized or declined more gradually (Fig. S5), and the contribution of anthropogenic emissions to ozone variability weakened considerably (Fig. S8).*

*Supplement:*

[Figure]

***Figure S5. Ground-based observed time series of $NO_2$, CO and $PM_{2.5}$.*** *Panels (a-c) show the summertime average time series of $NO_2$, CO, and $PM_{2.5}$ for China's five major city clusters from 2015 to 2023. Panels (e-f) present the annual average time series of $NO_2$, CO, and $PM_{2.5}$ for China's five major city clusters from 2015 to 2023.*

Line 249-251: The author needs to explain why the PRD is rising and why it is falling in other regions.

**Response:** Thank you for your suggestion. We have added a more detailed explanation, the related

modification is shown as follows:

*The most pronounced increases occurred in the FWP and BTH ($45.0 \pm 2.0\ \mu g\ m^{-3}$ and $42.1 \pm 2.0$ $\mu g\ m^{-3}$, respectively), whereas the PRD exhibited a relatively modest enhancement ($13.4 \pm 1.6\ \mu g\ m^{-3}$), reflecting its predominantly $NO_x$-limited photochemical regime versus VOC-limited regimes in other regions (Ren et al., 2022).*

Line 260: Authors should show a map to tell reader where is northern China?

**Response:** Thank you for your suggestion. The reference to "northern China" specifically pertains to the BTH (Beijing-Tianjin-Hebei) and FWP (Fenwei Plain) city clusters in northern China. We have now clarified this in the manuscript.

Line 260-262: Prior to 2017, PRD consistently recorded the lowest ozone concentrations among these major regions. Intuitively, the PRD appears to have the least potential for ozone reduction. Yet why did both the YRD and SCB regions exhibit smaller reduction margins than the PRD?

**Response:** Thank you for your comment. During phase II, MDA8 ozone decreased by $10.5 \pm 2.0$ $\mu g\ m^{-3}$ in BTH and $10.4 \pm 3.0\ \mu g\ m^{-3}$ in FWP, with smaller declines in YRD ($-4.8 \pm 3.8\ \mu g\ m^{-3}$), SCB ($-2.8 \pm 2.4\ \mu g\ m^{-3}$), and PRD ($-6.6 \pm 1.4\ \mu g\ m^{-3}$). These changes are attributed to anthropogenic emission controls. Our results are generally consistent with those obtained by Wang et al. (2023) using statistical methods. Notably, the PRD region showed relatively larger changes compared to YRD and SCB, which can be attributed to shifts in the ozone production sensitivity regimes. We have provided a detailed explanation of these changes in the main text:

*The SCB region consistently exhibited strong $NO_x$ limitation (>75%), whereas the PRD showed a gradual expansion of the transitional regime alongside a modest contraction of VOC-limited areas. These shifts in photochemical sensitivity correspond well with the ozone decrease observed during Phase II emission reductions.*

[Figure]

*Figure 6. Trends in the distributions of ozone production sensitivity regimes. Fractions of VOC-limited, NOx-limited, and transitional ozone sensitivity regimes across five key regions during the summertime (June to August) from 2018 to 2023, based on the FNR analysis. Panel (f) presents the overall trends for all five regions.*

Line 266-268: The author should inform readers: How much have anthropogenic emissions decreased in major regions due to the nationwide lockdown?

**Response:** Thank you for your suggestion. We have added quantitative information on emission reductions during the nationwide lockdown. The revised text reads:

*The COVID-19 pandemic (January-April 2020) introduced an unprecedented perturbation to anthropogenic activity, leading to sharp declines in industrial production, energy consumption, and transportation (Shi and Brasseur, 2020; Zheng et al., 2021). National emissions of $SO_2$, $NO_x$, $PM_{2.5}$, and VOCs were estimated to have decreased by 0.37 Tg (12%), 0.87 Tg (13%), 0.25 Tg (10%), and 1.07 Tg (12%), respectively, relative to the same period in 2019 (Geng et al., 2024).*

Line 273-275: The author suggests that the rise in ozone concentrations is due to suppressed NO titration. Therefore, the author could fully utilize TROPOMI data to analyze changes in sensitive zones during the COVID-19 period. This may help explain why ozone concentrations in the remaining 20% of cities decreased during the COVID-19 pandemic.

**Response:** Thank you for your suggestion. We applied the formaldehyde-to-$NO_2$ ratio (FNR)

diagnostic to classify ozone formation regimes and evaluate their spatial changes during the lockdown. The new analysis reveals clear regional differences: the North China Plain (NCP) and Yangtze River Delta (YRD) shifted toward more VOC-limited conditions due to sharp $NO_x$ reductions, while parts of southern China remained in $NO_x$-limited or transition regimes. The updated manuscript includes corresponding maps to illustrate these spatial patterns (Fig. S13). the related modification is shown as follows:

*Spatial distributions of ozone formation sensitivity during the COVID-19 lockdown (Fig. S13) reveal that most of China was in a transitional regime, with major urban clusters remaining VOC-limited and only limited areas in southern China being $NO_x$-limited. This spatial pattern aligns with the observed widespread ozone increases during the lockdown (Fig. S7).*
*Supplement:*

[Figure]

**Figure S13. Ozone formation sensitivity regimes during COVID-19.** *Spatial distribution of ozone formation sensitivity regimes in China from January to April 2020 during the COVID-19 pandemic. The hollow triangles represent the geographical coordinates of five key urban clusters in China.*

Line 275-276: The author should explain how anthropogenic emissions have changed during this period.

**Response:** Thank you for your suggestion. We have added Ground-based observed time series of $NO_2$, CO and $PM_{2.5}$, which provide insights into the changes in anthropogenic emissions during this period. The related modification is shown as follows:

*In the post-pandemic period (2020–2023), concentrations of $NO_2$, CO, and $PM_{2.5}$ stabilized or declined more gradually (Fig. S5), and the contribution of anthropogenic emissions to ozone variability weakened considerably (Fig. S8).*
*Supplement:*

[Figure]

***Figure S5. Ground-based observed time series of NO₂, CO and PM₂.₅.*** *Panels (a-c) show the summertime average time series of NO₂, CO, and PM₂.₅ for China's five major city clusters from 2015 to 2023. Panels (e-f) present the annual average time series of NO₂, CO, and PM₂.₅ for China's five major city clusters from 2015 to 2023.*

Line 295: Where is North China Plain?

**Response:** Thank you for your comment. We have marked the coordinates of the North China Plain (NCP) on the map. The related modification is shown as follows:

*Supplement:*

[Figure]

***Figure S10. Spatial and temporal variations of satellite NO₂.*** *Map of average levels of satellite-observed NO₂ from June-August 2018 to 2023. The rectangle in panel (a) represents the extent of the North China Plain (NCP).*

Line 297: "The Yangtze River Delta" to "YRD".?

**Response:** Thank you for your suggestion. We have corrected the error.

Line 298-299: You can show a Δtemperature map from ERA5 data.

**Response:** Thank you for your suggestion. We have added a spatial distribution of daytime temperature differences between HW and NHW. The related modification is shown as follows:

*Between 2018 and 2023, NO₂ columns over the North China Plain (NCP) declined from 4.13 × 10¹⁵ to 3.85 × 10¹⁵ molecules cm⁻², while HCHO remained stable until 2021, followed by a sharp increase in 2022. The spatial pattern of temperature anomalies between heatwave (HW) and non-heatwave (NHW) periods (Fig. S11) reveals strong positive differences in the YRD and SCB, consistent with enhanced biogenic and anthropogenic VOC emissions under extreme heat (Qin et al., 2025; Tao et al., 2024).*

*Supplement:*

[Figure]

**Figure S11. Spatial distribution of daytime (11:00-17:00) temperature differences between HW and NHW.** *The HW period is defined as July 16 to August 31, 2022, while the corresponding period in other years is considered as NHW.*

Line 306-307: The author might use this to explain why the PRD's changes differ from those of other cities in Section 3.2.

**Response:** Thank you for your suggestion. We have incorporated the concept of differences in ozone sensitivity intervals in Section 3.2 to clarify why the changes in the PRD differ from those of other cities. The related modification is shown as follows:

*The most pronounced increases occurred in the FWP and BTH (45.0 ± 2.0 μg m⁻³ and 42.1 ± 2.0 μg m⁻³, respectively), whereas the PRD exhibited a relatively modest enhancement (13.4 ± 1.6 μg m⁻³), reflecting its predominantly NOₓ-limited photochemical regime versus VOC-limited regimes in other regions (Ren et al., 2022).*

*The SCB region consistently exhibited strong NOₓ limitation (>75%), whereas the PRD showed*

*a gradual expansion of the transitional regime alongside a modest contraction of VOC-limited areas. These shifts in photochemical sensitivity correspond well with the ozone decrease observed during Phase II emission reductions.*

Figure 3: It is recommended that the author label the locations of major regions on this map.

**Response:** Thank you for your suggestion. In the revised manuscript, we have replaced the corresponding image with a map showing the trend of ozone sensitivity intervals across the five major city clusters of China from 2018 to 2023. Additionally, the map in the supplementary material has also been updated to reflect the latitude and longitude range corresponding to the five major city clusters.

**The related modification is shown as Fig. 6 and Fig. S12.**

Section 3.3: Lack of quantitative results. Figure 3 fails to clearly show the interannual changes in sensitive areas. The authors should adopt a more explicit presentation method to highlight the differences in area among sensitive zones in each region.

**Response:** Thank you for your insightful comments and suggestions. We have revised Section 3.3 and enhanced the presentation of Figure to better highlight the interannual changes in sensitive areas. A more explicit method has been adopted to clearly show the differences in area among the sensitive zones in each region. The related modification is shown as follows:

*To diagnose the evolving chemical sensitivity of ozone production, we examined the spatiotemporal evolution of the HCHO/NO$_2$ ratio (Text S1). Figure S12 shows that this ratio exhibited regionally distinct transitions from 2018 to 2023, reflecting shifts in photochemical regimes. Figure 6 summarizes the relative contributions of VOC-limited, NO$_x$-limited, and transitional regimes across the five key regions. In BTH, NO$_x$-limited areas accounted for ~82% of the domain in 2018 and remained above 45% thereafter, while VOC-limited regions declined from ~14% to ~2%. In FWP, summer ozone formation was largely controlled by NO$_x$-limited and transitional regimes. The YRD underwent a notable shift from VOC- to NO$_x$-limited chemistry, with VOC-limited fractions decreasing from ~35% in 2018 to ~22% in 2023, particularly during 2022 when extreme heat amplified VOC emissions and photochemical activity (Qin et al., 2025; Tao et al., 2024). The SCB region consistently exhibited strong NO$_x$ limitation (>75%), whereas the PRD showed a gradual expansion of the transitional regime alongside a modest contraction of VOC-limited areas. These shifts in photochemical sensitivity correspond well with the ozone decrease observed during Phase II emission reductions.*

[Figure]

*Figure 6. Trends in the distributions of ozone production sensitivity regimes. Fractions of VOC-limited, $NO_x$-limited, and transitional ozone sensitivity regimes across five key regions during the summertime (June to August) from 2018 to 2023, based on the FNR analysis. Panel (f) presents the overall trends for all five regions.*

Line 337-350: Authors cannot merely provide quantitative descriptions; ACP journals require authors to conduct deeper analysis of these quantitative results.

**Response:** Thank you for your insightful comments and suggestions. We have revised the manuscript to provide a deeper analysis of the quantitative results presented. This includes a more detailed interpretation of the data and its implications.

**The corresponding modifications can be found in Section 3.3.**

Line 356-357: That's a boring sentence. Although the author provides some explanation in lines 358-359, this explanation is overly broad. For example, why are RH and temperature the dominant factors in other regions, while shortwave radiation and RH are the dominant factors in the YRD region? What is the relationship between temperature and radiation? Which variable is the fundamental cause of ozone concentration changes, and what role does RH play in this process?

**Response:** Thank you for your insightful comments. We have removed the sentence and we have expanded our analysis and discussion to clarify the distinct roles of temperature, humidity, and solar

radiation in ozone formation across different regions. Our extended machine learning analysis (see Fig. S16) reveals region-specific sensitivities of ozone to these meteorological factors. The related modification is shown as follows:

*Partial dependence analysis (Fig. S16) further illustrates the nonlinear responses of ozone to key meteorological factors (T, RH, SR) for representative cities in each cluster, revealing clear regional contrasts. In Beijing (BTH), ozone concentrations show the strongest positive response to T (Fig. S16a), consistent with the enhancement of reaction kinetics and biogenic VOC emissions under hot conditions. This behavior reflects the thermodynamic coupling between surface heating, boundary-layer expansion, and photochemical production. In Nanjing (YRD), ozone is more sensitive to solar radiation than to temperature (Fig. S16c), highlighting the dominant role of actinic flux in controlling radical production during warm and dry conditions. Yang et al. (2024) similarly reported that high-temperature and low-RH conditions over the NCP and YRD enhance photochemical ozone formation, with chemical production being the dominant process driving ozone buildup during the most polluted months. In the SCB, both T and RH exhibit strong influences, while in the PRD, ozone variability is shaped primarily by T and large-scale circulation patterns associated with subtropical maritime flow and typhoon incursions from the Northwest Pacific (Chen et al., 2024; Wang et al., 2024a; Wang et al., 2022a).*

[Figure]

**Figure S16. Partial dependence of ozone on T, RH, and SR for representative cities.** *Panels show the 3D-dependence plots of MDA8 ozone with RH, T, and SR for representative cities in BTH, FWP, YRD, and PRD, including MDA8 ozone-RH-T, MDA8 ozone-RH-SR, and MDA8 ozone-SR-T.*

Line 369-371: Figure S13 shows that the correlation between T and SR in inland southern China is very low during both HW and NHW periods. Why is this? Additionally, the correlation for RH between HW and NHW appears significantly different. I recommend that the authors adjust the colorbar in Figure S13E-F to a red-blue scale with zero centered. This may reveal positive correlations for RH in certain regions.

**Response:** Thank you for your insightful comments and suggestions. We appreciate the reviewer pointing out the issue. The correlation does not imply causality, and Figure S13 should not have been used as evidence. We have removed the corresponding content from the manuscript.

Line 379: "amplifying" to "increasing".

**Response:** Thank you for your suggestion. We have replaced "amplifying" with "increasing".

Line 383-384: I do not agree that. Cloud cover directly influences temperature and shortwave radiation variations and should not be considered a secondary effect.

**Response:** Thank you for your suggestion, we agree with you. During extreme weather events, ozone variation is influenced by multiple meteorological parameters, and there is no primary or secondary relationship among them. We have revised the manuscript accordingly. The revised sentence is as follows:

*A multi-year comparison (Fig. 7) highlights the opposing effects of key meteorological variables – including RH, T, boundary layer height (BLH), total precipitation (TP), and surface pressure (SP) – on MDA8 ozone.*

Line 387: What is the RH effect?

**Response:** Thank you for your comment. We have modified the corresponding text to avoid ambiguity. The revised sentence is as follows:

*The trend in $\triangle$SHAP values under high-humidity conditions from 2015 to 2023 (Fig. S20) further confirms the model's ability to capture the suppressive effects of wet weather conditions on ozone formation.*

---

## Author Response (AR2)

Dear Editor,

Thank you very much for your and the reviewers' thoughtful and constructive comments on our manuscript. We have carefully addressed all comments and suggestions point by point and have revised the manuscript accordingly. Detailed responses to the reviewers' comments are provided below (*in blue*), and all corresponding revisions are highlighted *in red* in the revised manuscript.

Thank you very much for your time and consideration!
We are looking for forward to hearing from you.

Sincerely,
Yunjiang Zhang, on behalf of all co-authors
Nanjing University of Information Science and Technology, Nanjing, China
Email address: yjzhang@nuist.edu.cn

**Response to the Referee**

The authors have satisfactorily addressed all of my previous comments and concerns. I also note that the responses to the other reviewers' comments are thorough and that appropriate revisions have been made accordingly. Overall, the manuscript is substantially improved, and I believe it is close to being suitable for publication after the authors address the following new minor issues:

**Response:** We sincerely appreciate the reviewer's positive and constructive feedback on our manuscript. Your suggestions have been invaluable in enhancing the quality and clarity of our work. We have thoroughly revised the manuscript in response to your comments and believe that these changes have significantly improved it. Thank you again for your insightful input.

Line 88: Please revise "machine learning-based framework" to "machine learning-based model framework."

**Response:** Revised.

Line 89: Replace "respective roles" with "relative contribution."

**Response:** Replaced.

Line 145: Replace "errors" with "uncertainty."

**Response:** Replaced.

Line 196: Please carefully evaluate the use of "significant" or "significantly" throughout the manuscript, especially where no significance test is provided. In addition, could the authors offer a numerical range for the uncertainty reduction here?

**Response:** Thank you for your insightful suggestions. We carefully reviewed the use of "significant" and "significantly" throughout the manuscript and confirmed that these terms were indeed used in contexts without significance testing. To more accurately convey our results, we have removed "significant" from the revised text and included specific descriptions of the uncertainty reduction. The modified content is as follows:

*Notably, inclusion of time-related variables could reduce model uncertainty compared to simulations excluding these predictors. The average uncertainty decreased by approximately 2–4% at the regional-mean level (Fig. S3).*

Line 200: The description should be improved. The authors may directly state that the x-axis represents the years used for model training, and the y-axis represents the years predicted by the trained model.

**Response:** Thank you for your suggestion. We have revised the description for clarity, stating that the x-axis represents the years used for model training, while the y-axis indicates the years predicted by the trained model. This clarification should enhance the reader's understanding of the figure.

[Figure]

*Figure 2. Uncertainty assessment of the FEA method. The uncertainty for the FEA method is calculated using the approach described in Text S2. The x-axis represents the years used for model training, and the y-axis represents the years predicted by the trained model. The diagonal line in each sub-panel represents the changes in the residuals of the models.*

Line 270: The abbreviations "BTH" and "YRD" should be defined upon first use. Also, Lines 278–279 contain redundant definitions, please remove duplicates.

**Response:** Thank you for your suggestion. We have defined the abbreviations "BTH" (Beijing-Tianjin-Hebei) and "YRD" (Yangtze River Delta) upon their first use in the manuscript in line 66. Additionally, we have removed the redundant definitions found in lines 278-279 to ensure clarity and conciseness. We also conducted a thorough review of the entire manuscript to ensure that all abbreviations and their corresponding full terms are used correctly and consistently.

Lines 289–291: I suggest removing this sentence, as a similar concluding remark has already been presented earlier.

**Response:** Thank you for your suggestion. We have removed it.

Line 301: Please replace "first phase" with "Phase I," and ensure consistent formatting for subsequent phases.

**Response:** Thank you for your suggestion. We have made the requested change by replacing "first phase" with "Phase I" and have ensured consistent formatting for all subsequent phases throughout the manuscript.

Line 361: I recommend removing the phrase "suggesting that ozone responses to further emission reductions may have reached a saturation point."

**Response:** Thank you for your suggestion. We have removed the phrase.

Line 399: Please clarify which region "f" refers to in this context.

**Response:** Thank you for your comment. We have clarified in the figure legend that "f" represents the overall conditions across the five regions, and we have added relevant annotations on the y-axis of "f" for further clarification.

[Figure]

*Figure 6. Trends in the distributions of ozone production sensitivity regimes. Fractions of VOC-limited, $NO_x$-limited, and transitional ozone sensitivity regimes across five key regions during the summertime (June to August) from 2018 to 2023, based on the FNR analysis. **a-e** the trend across the five city cluster regions in China during the summer months (June, July, and August): BTH, FWP, YRD, SCB, and PRD, respectively. **f** presents the overall trends for all five regions.*

Line 418: It would be more accurate to state that the analysis "provides an indication of meteorological conditions" rather than drawing a direct conclusion.

**Response:** Thank you for your suggestion. We have revised the sentence for greater accuracy. The modified content is as follows:

*The following summer (2023) featured anomalously heavy rainfall, resulting in sharp ozone suppression (–17.8 ± 2.3 µg m$^{-3}$ in the YRD and –9.7 ± 3.3 µg m$^{-3}$ in the SCB). This reduction coincided with a remarkable increase in precipitation, i.e., 102% in YRD and 35% in SCB (Fig. S14), indicating that rainy meteorological conditions may have suppressed ozone production.*

Lines 436–438: This sentence can be further streamlined for clarity.

**Response:** Thank you for your suggestion. We have streamlined the sentence for clarity. This modification improves the clarity of the statement while retaining its original meaning. The modified content is as follows:

*Consistent with these findings, Yang et al. (2024) reported that high-temperature and low-RH conditions over the NCP and YRD could enhance photochemical ozone formation, with chemical production dominating during peak pollution periods.*

Line 463: Please revise the section title to "Reshaping distributions of ozone by climate change and emission controls."

**Response:** Revised.

Line 477: A brief clarification of the scenarios referenced here would be helpful, as the current wording is vague even though earlier sections describe them in detail.

**Response:** Thank you for your suggestion. To improve clarity, we have added a brief clarification of the scenarios referenced in Line 477. The modified content is as follows:

*Three sensitivity simulations (see Section 2.5 and Fig. S21) confirm this robustness: trend slopes range from 0.11–0.14 µg m$^{-3}$ yr$^{-1}$ in BaseBTH (high-pollution scenario), 0.05–0.10 µg m$^{-3}$ yr$^{-1}$ in the BaseYRD (moderate-pollution scenario), and 0.03–0.10 µg m$^{-3}$ yr$^{-1}$ in the BasePRD (low-pollution scenario).*

Line 525: Please check whether "anthropogenic precursor emissions" refers specifically to HCHO here and revise accordingly for accuracy.

**Response:** We thank the reviewer for raising this point. In the original text, anthropogenic precursor emissions was not intended to refer only to HCHO, but rather to the combined influence of multiple anthropogenic ozone precursors, including $NO_x$, CO, and anthropogenic VOCs. To avoid potential ambiguity, we have revised the sentence to explicitly clarify this broader definition. The revised text now reads:

*Our results revealed that increased anthropogenic emissions were the dominant driver of the sharp rise in summertime MDA8 ozone concentrations during the Phase I, contributing an average increase of $23.2 \pm 1.1$ $\mu g$ $m^{-3}$.*

Line 538: I suggest removing the word "growing."

**Response:** Removed.

Lines 550–551: The conclusion should be reframed to indicate that a warming climate modulates the long-term evolution of ozone trends. The subsequent sentence may then be adjusted for a smoother logical transition.

**Response:** Thank you for your insightful suggestion. We have reframed the conclusion to indicate that a warming climate modulates the long-term evolution of ozone trends, and we adjusted the subsequent sentence for a smoother logical transition. The modified content is as follows:

*Good correlations between the modelled ozone and surface temperature (r = 0.72-0.93) across major urban clusters indicated that climate warming exerts a persistent control on the long-term evolution of ozone. While reductions in precursor emissions have improved ozone control efficiency, the direct enhancement of ozone by rising temperatures increasingly interferes with, and in some regions may partially offset, the air-quality benefits achieved through emission mitigation. Together, these findings highlight that effective ozone management in a warming world will require integrated strategies that jointly address emission reductions and climate adaptation.*